# Trend analysis and prediction of fabric tear performance testing processes based on the BLTT-FT model

Qingchun Jiao[1], Yifan Zhang[1], Yang Lu[2]*, Bo He[2], Min Zhu[2], Kuokuo Wang[1]

**1** School of Automation and Electrical Engineering, Zhejiang University of Science and Technology, Hangzhou, China, **2** Zhejiang Light Industrial Products Inspection and Research Institute, Hangzhou, China

* 13857154948@163.com

## Abstract

Fabric tearing performance testing experiment is an important part of evaluating fabric durability. The aim of this paper is to solve the problem of real-time prediction of fabric tearing performance testing by effectively extracting key features from experimental data and constructing a prediction model applicable to the process of fabric tearing performance testing. In this study, the trend prediction model for the experimental process of fabric tear performance testing (BLTT-FT) based on the "bidirectional long- and short-term attention mechanism" is adopted. A prediction model combining the improved Bi-directional Long Short-Term Memory (BiLSTM) structure, Transformer encoding layer, and Temporal Convolutional Network (TCN) layer is proposed. While considering sequence information globally, the model captures the bidirectional dependence of time series, reduces model complexity through the TCN layer, and finally optimizes prediction accuracy via the fully connected layer and activation function, thus achieving multi-step prediction. Analysis of variance (ANOVA) indicates that, across multiple datasets constructed from fabrics with different elasticity grades, the model shows extremely significant differences ($p < 0.001$) in the metrics of Mean Absolute Error (MAE), Root Mean Square Error (RMSE), and Mean Absolute Percentage Error (MAPE) at each prediction step. Furthermore, it maintains a low error level even in the long-range prediction scope: the average RMSE of multi-step prediction is 0.0881, the average MAE of multi-step prediction is 0.0609, the average MAPE of multi-step prediction is as low as 3.06%, and the average coefficient of determination ($R^2$) of multi-step prediction is as high as 0.9572. The ablation experiments confirm that multi-modular hierarchical modeling effectively solves the problem of detail accuracy of single-step prediction and long-range dependence of multi-step prediction. The results show that the proposed model performs well in real-time trend prediction results for different data sets constructed from fabrics with different elasticity grades. By predicting the dynamics of the experimental process

**Data availability statement:** The data supporting the results of this study have been stored on the Github website at: https://github.com/yuyuyu123YUYUYU/DS.

**Funding:** This study was funded by the following grants: 1. Research and Application of Key AI Evaluation Technologies for Fabric Pilling Based on Machine Vision/Zhejiang Provincial Market Supervision Administration (Project No. ZD2025007). 2. Research and Establishment of Data Identification Model for Abnormal Behaviors in Inspection and Testing/Open Project of the National Market Regulation Technology Innovation Center (Digital Research and Application of Market Regulation) (Project No. 2024SF02WX0007). 3. Science and Technology Planned Project of the State Administration for Market Regulation (Project No. CY2023213). 4. "Chu Ying" Project (Core Project) of Zhejiang Administration for Market Supervision (2022MK057). 5. Natural Science Foundation of Zhejiang Province (Project No. LGG20F020008).

**Competing interests:** The authors have declared that no competing interests exist.

of fabric tearing performance testing in real time, this study has exploratory value in improving the experimental efficiency and optimizing the experimental process.

## Introduction

With the development of modern industrial technology, elastic fabrics have been widely used in many fields such as clothing and medical treatment by virtue of their excellent elasticity and comfort [1,2]. The types of fabric structures are categorized into four types: woven fabrics, braided fabrics, tertiary fabrics, and nonwovens. Compared with other fabrics, woven fabrics have good dimensional stability and the highest covering yarn stacking density in the warp and weft directions. Whenever a new fabric is produced, a series of tests such as tearing performance tests, pilling, tensile strength, etc., are usually carried out, which play a vital role in assessing the quality of the product [3]. The samples are tested and analyzed, and if the sample passes all tests and meets the requirements, the fabric is ready for mass production. The tearing performance of textiles is an important analytical judgment index for evaluating the performance of clothing [4], and the tearing phenomenon is that the yarns in the fabric break in sequence until the fabric is completely torn, which is closer to the situation of sudden rupture in the actual use, and reflects the toughness performance of textiles more effectively.

With the continuous progress of cutting-edge technologies such as sensors, artificial intelligence, big data, and the industrial Internet of Things, the traditional manufacturing industry is accelerating its transformation to smart manufacturing [5]. Scholars at home and abroad have carried out extensive exploration of various industrial experimental processes, especially focusing on the innovation and development of information acquisition means, based on which, the experimental monitoring technology can be mainly divided into two categories: direct monitoring and indirect monitoring [6]. Traditional mechanical testing methods are mainly used to measure and evaluate the tear strength of elastic fabrics, including pendulum, single tongue, or trapezoidal specimen tear tests, etc. These methods obtain tear strength data by applying force at a fixed rate until the fabric undergoes tearing [7], and although the direct monitoring methods have played a role in the past, there are limitations in their ability to achieve real-time online tracking and analysis of the data. In the context of today's scientific research and Industry 4.0 era, with the deep integration of sensor technology, artificial intelligence, big data analysis, and Internet of Things (IoT) technology, the traditional mode of experimental monitoring and analysis is gradually shifting to the direction of intelligence and predictability. Indirect monitoring technology continuously captures multivariate physical signals in the experimental process by integrating a high-precision sensor network [8] and utilizes machine learning and deep learning algorithms to excavate the hidden laws behind the physical signals and the complex correlation between the experimental process, a strategy that lays a solid data foundation for intelligent prediction [9]. Sensors have been deployed in the test environment in some existing studies to indirectly acquire test data and analyze the trend of the experimental process in order to achieve the effect of tracing and predicting the experimental process [10]. Kuntoğlu M et al. [11] systematically

analyzed the correlation between sensor data and tool wear and developed a multi-sensor tool condition monitoring system to effectively identify the wear state of the tool by monitoring the type of energy used in the cutting process, thus enabling real-time monitoring of the cutting process. Pazikadin AR et al. [12] used an artificial neural network to predict solar power generation data from data measured by a solar irradiance sensor.

When this type of activity is performed in a fabric performance testing laboratory, it is susceptible to factors such as equipment operation, personnel handling, and test environment, and lacks non-intrusive system awareness. In the preliminary stage, we proposed a multi-source data-driven state perception and classification study for fabric tearing performance detection [13], which solved the problem that the experimental results may be affected by the operator's personal experience and subjective judgments through the designed non-intrusive system [14], and ensured the consistency and accuracy of the judgments of each state transition, as well as carried out the indirect monitoring of the electricity consumption. The value of electrical power as an indicator of the operating status of equipment is manifested at several levels [15], especially in the fields of industrial automation, equipment monitoring, and fault diagnosis.

In textile tearing performance testing equipment, the electric power parameter reflects the real-time status and potential problems of the equipment operation. The electrical power reflects the energy consumption of the equipment during operation, with different work corresponding to different energy consumption patterns. Monitoring the changes of electric power parameters in the experimental process in real-time provides the experimenter with immediate decision support [16] to optimize the experimental process, and improve the experimental efficiency and the reliability of the results. In a previous study of the comparison between electrical power parameters and mechanical characteristics, it was found that the fluctuation trends of the two were consistent, proving the idea that mechanical variations can be represented with less power-aware data, but predictive studies are still lacking in their basis. This study is dedicated to bridging this research gap and seeking a more effective state prediction method for further optimization of the experimental process.

The research in this paper includes the following aspects:

(1) In this paper, we propose an experimental process trend prediction model (BLTT-FT) based on the "Bidirectional Long and Short-Term Attention Mechanism" for fabric tearing performance testing. An innovative prediction framework is formed by combining Transformer, Temporal Convolutional Neural Network (TCN) and improved BiLSTM.

(2) The proposed model combines a BiLSTM structure composed of improved LSTMs, a Transformer encoding layer, and a TCN layer. It utilizes the improved Bi-LSTM to capture the bidirectional dependence of sequences, employs the self-attention mechanism of the Transformer encoding layer to consider sequence information globally, and then adopts the TCN layer to optimize the processing of variable-length sequences so as to reduce model complexity. Finally, the prediction accuracy is optimized through the fully connected layer and activation function.

(3) The proposed model is utilized for multi-step prediction of power changes, and the effectiveness and accuracy of the model in prediction are verified through comparative analysis with other models, and the synergy of the model components is remarkable, especially when dealing with the tearing experimental data of fabrics with different elasticity levels, which demonstrates excellent prediction performance and strong generalization ability. The model cleverly combines with the previous situational awareness system, and makes full use of the electric power parameter to predict the trend of the equipment during the fabric tearing experiments.

In summary, the aim of this study is to develop a BLTT-FT model based on electrical power sequences for application in real-time prediction of fabric tearing experimental processes. By carefully analyzing the mapping relationship between the signal characteristics of the electric power sequence and the whole experimental process, on the basis of the historical electric power data collected in the previous period, the existing functions of the situational awareness system are improved and perfected, and the overall prediction function for the experimental process of fabric tearing is added, so as to improve the efficiency and accuracy of the experiments and to reduce the experimental processes to a certain extent and optimize the whole experimental process.

## Related work

### Application of artificial intelligence in fabric performance prediction

In recent years, more and more researchers have begun to adopt data-driven prediction methods to establish prediction models based on the static attributes of woven fabrics (e.g., fiber type, yarn specification, fabric structure, etc.), and have achieved a certain degree of prediction accuracy.

Ahirwar M et al. [17] developed a machine learning-based neural network approach to predict the performance of fabric using fabric parameters as input and warp and weft tear strength as output. HOSSAIN MM et al. [18] used a correlation regression model to explain the effect of structural parameters on the tear and tensile strength of various base fabric designs to develop a neural network model for predicting tear and tensile strength. Ribeiro R et al. [19] successfully developed models with high prediction accuracy by applying machine learning techniques and analyzing multiple features of the textile production process. Xiao Q et al. [20] proposed an intelligent ball-starting prediction model based on the BP neural network and an optimization model based on a genetic algorithm to improve the training speed and accuracy of the ball-starting prediction. Tu Y F et al. [21] conducted a systematic review of AI-driven fabric performance and handle prediction technologies (focusing on model mechanisms, dataset diversity, and prediction accuracy). They identified research gaps and challenges in this field, providing practical references for improving AI prediction capabilities and guiding future innovations in textile technology. Sarkar J et al. [22] used Adaptive Neuro-Fuzzy Inference System (ANFIS) and Artificial Neural Network (ANN) methods to develop predictive models for textile substrate absorption properties, which can help in the scale-up of functional textiles. Doran EC et al. [23] proposed artificial neural network (ANN) and support vector machine (SVM) models to predict the quality characteristics of cotton/elastane fiber core yarns using fiber quality and spinning parameters.

Most of the existing research focuses on the prediction of finished product quality, while the real-time monitoring and prediction of the fabric experimental process itself is rarely involved. In addition, most of the existing monitoring techniques focus on data analysis under static conditions and lack a real-time response mechanism to dynamic changes in the experiment, which cannot meet the needs of the modern textile industry for efficient, accurate, and intelligent production processes.

### Predictive applications of time series data

Since the data during the fabric performance experiments are all time-series data, the methods for time-series data prediction are broadly categorized into three types: statistical analysis, machine learning, and deep learning. Guo N et al. [24] created a hybrid prediction model by combining ARIMA (Autoregressive Integrated Moving Average) with SVR (Support Vector Regression) to more accurately predict electricity consumption data collected in an Internet of Things (IoT) environment. Xie Y et al. [25] proposed a hybrid model of ARIMA and triple exponential smoothing, which can accurately predict linear and nonlinear relationships in container resource loading sequences. However both ARIMA and exponential smoothing methods rely heavily on historical data, and they are not the right choice for forecasting long-term time series if there is a lot of variability in the data.

In recent years, deep learning has achieved excellent results in time series data prediction. BiLSTM is able to better capture the contextual information when dealing with sequential data: Guo Y et al. [26] proposed an MES combined load prediction method based on bi-directional long short-term memory (BiLSTM) multi-task learning and achieved good prediction results; Wu K et al. [27] proposed a hybrid prediction model based on wavelet threshold denoising (WTD), variational modal decomposition (VMD), and bi-directional long and short-term memory (BiLSTM) networks in order to reduce the short-term household load forecasting errors due to small load sizes and different residential electricity consumption behaviors, which provided more stable and accurate predictions under trend feature extraction, and improved the short-term household load forecasting accuracy. Transformer effectively captures long-term dependencies in time-series data, ensuring chronological accuracy, and its flexible model structure allows for adjustments to accommodate data of varying

complexity, while the encoder-decoder architecture is particularly well suited for predicting future points in time [28]: Qu K et al. [29] applied the Transformer model in Natural Language Processing (NLP) to the field of wind power prediction, which not only can accurately extract different correlation levels between multiple wind farms but also can give accurate wind power prediction results; Reza, S. et al. [30] designed a multi-attention based transformer model for traffic flow prediction using five identical encoder and decoder layers and combined a comparative analysis between gated recurrent units and a long term memory based model with good performance in effectively predicting long term traffic flow patterns. Guo J et al. [31] proposed a hybrid method for bearing failure prediction: constructing health indicators through CEEMDAN and KPCA, extracting multi-domain features using dual-channel Transformer with CBAM, and realizing RUL probability prediction by combining the $3\sigma$ criterion and Wiener's process, which verified the time-series modeling potential of Transformer. In recent years, temporal convolutional neural networks (TCNs) have emerged as a new approach for dealing with temporal problems, with significant advantages in weight sharing and convolutional local perception, and have achieved excellent performance in fusion prediction with other models: Lu P et al. [32] used TCN to extract the hidden temporal features in wind power data to establish the Informer wind power prediction model; Liu S et al. [33] and others proposed parallel structure TCN-LSTM wind power prediction model based on Savitzky-Golay filtering and TCN; Zhang G et al. [34] proposed a novel hybrid model based on adaptive quadratic decomposition method and robust time convolution network (RTCN) for wind speed prediction.

This paper will extend the applicability of these models to a wider range of complex prediction tasks, including dynamic correlation and trend prediction of electrical power sequences and experimental processes for fabric tear performance testing.

# Research methods

## Overall model design

The model developed in this paper predicts the trend of electric power parameters monitored during the experimental process of fabric tearing performance, and the overall work consists of the following three parts: data preprocessing part, data division part, and BLTT-FT part. The main flow is shown in Fig 1.

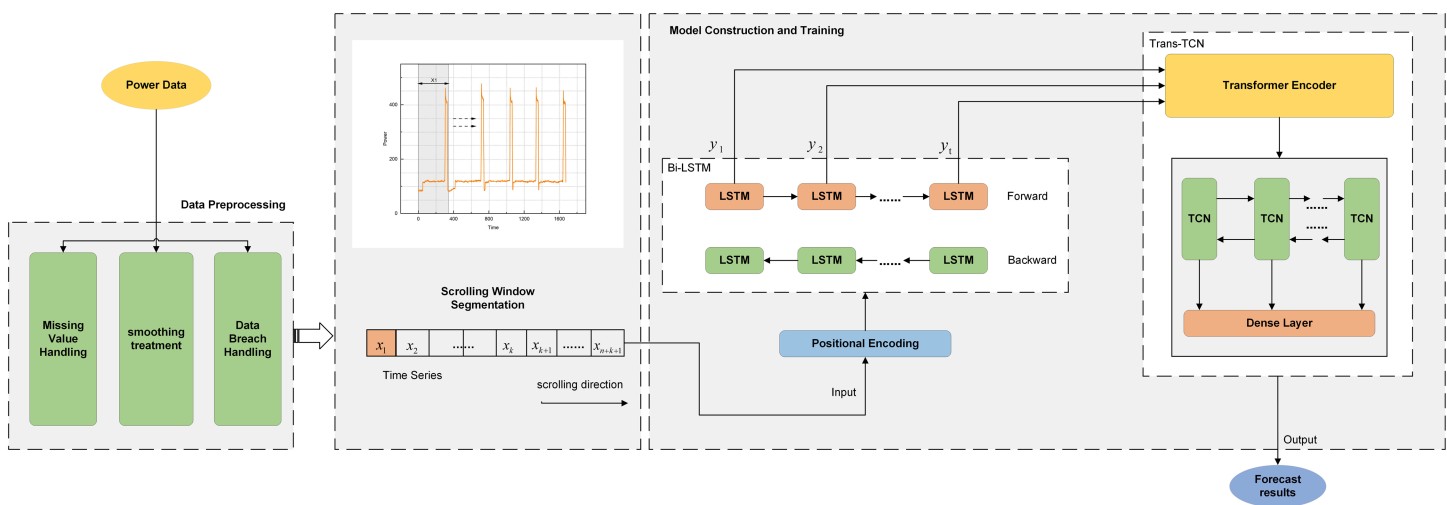

**Fig 1**. **Main flowchart of the overall model design.** Flow of electrical parameters prediction for the experimental process of fabric tearing performance based on BLTT-FT fusion model.

The model first receives the raw data stored by the sensors in the database and composes the complete sample data by taking the electric power parameters in the order of the moments when the experiment is performed. Then the data is preprocessed, mainly for missing values, smoothing, and data leakage. Data partitioning presents an innovative integration of the principles of discrete-time state machines and the concept of dynamically tuned rolling windows, focusing on the prediction of multi-step states occurring after a single "change- tensile -reset" state. The model is trained on the divided data, and then gradually rolls forward. Whenever the "change- tensile-reset" state is completed once, the state window will roll to the next state point, and the BLTT-FT prediction model will be updated accordingly, forming a closed loop of continuous iteration and optimization. In this way, the real-time prediction environment is simulated and the output prediction results are obtained.

## BLTT-FT system network construction

In this paper, the Transformer model is used as the basis, and a temporal convolutional network (TCN) is introduced to modify it so that it is more suitable for the prediction of time-series data in this paper. The Transformer model allows for parallel computing and good access to global information, however, the ability to capture global information is weak and not ideal for direct use as a predictor. TCN captures both high-level and low-level features with stable gradients and enables the model to process time series information in parallel, improving the prediction accuracy and training efficiency of the model. The improved Transformer model can fully utilize the advantages of Transformer and TCN to better predict the sequence data, and the improved model is called the "Trans-TCN" network structure.

However, the "Trans-TCN" network structure lacks sequential information, and although its position encoding uses sine and cosine to model positions, it is not sufficient for complete modeling and suffers from a certain lack of information. Therefore, the data will be processed using a bi-directional long and short-term memory network to capture the sequential dependencies before transferring them to the "Trans-TCN" polytope self-attention point. Due to the large sample size of the dataset, this paper will use the BiLSTM network model composed of improved LSTM to speed up the data learning speed, and the improved BiLSTM neural network model is called "Bi-LSTM" network architecture. Therefore, combining it with "Trans-TCN" can improve the prediction efficiency of the model.

In summary, after the input data is processed by the positional coding layer, the sequence features are captured by the "Bi-LSTM" and then input to the "Trans-TCN". Specifically, it is first processed by the encoder of the Transformer, then the features are further extracted by the TCN layer, and finally, the full connectivity layer and activation function are utilized for dimensionality reduction. The main structure consists of a position coding layer, a Bi-LSTM layer, a Transformer encoder layer, a TCN layer, and a fully connected layer, and the overall network structure is called "BLTT-FT" as shown in Fig 2.

## Trans-TCN network construction

This section describes in detail the improvement of TCN on the Transformer model, and the improved structure is called the "Trans-TCN" network structure. Although the Transformer model was originally designed for machine translation, it can still be applied to time series prediction by exploiting its architectural potential. Therefore, its encoder part is used as the basic model. The "Trans-TCN" network structure is shown in Fig 3.

(1) The Transformer encoder consists of multiple encoder layers, each consisting of two sub-layer blocks. The first sub-layer block includes a multi-head attention layer and a normalization layer connection; the second sublayer block includes a feed-forward layer and a normalization layer connection. The core part of the coding layer is the multi-head attention mechanism, and this part is the key part of the Trans-TCN network structure to realize accurate prediction.

A multiple attention strength mechanism consists of multiple heads with self-attention. The self-attention process in the multi-head attention mechanism can be described as a process solved by a query vector and a set of key-value vector matrices. Where the query vector ($Q$), key vector ($K$), and weight vector ($V$) are transformed from the previous set of outputs.In the actual operation, the model simultaneously computes the attention functions on a set of query vectors packs

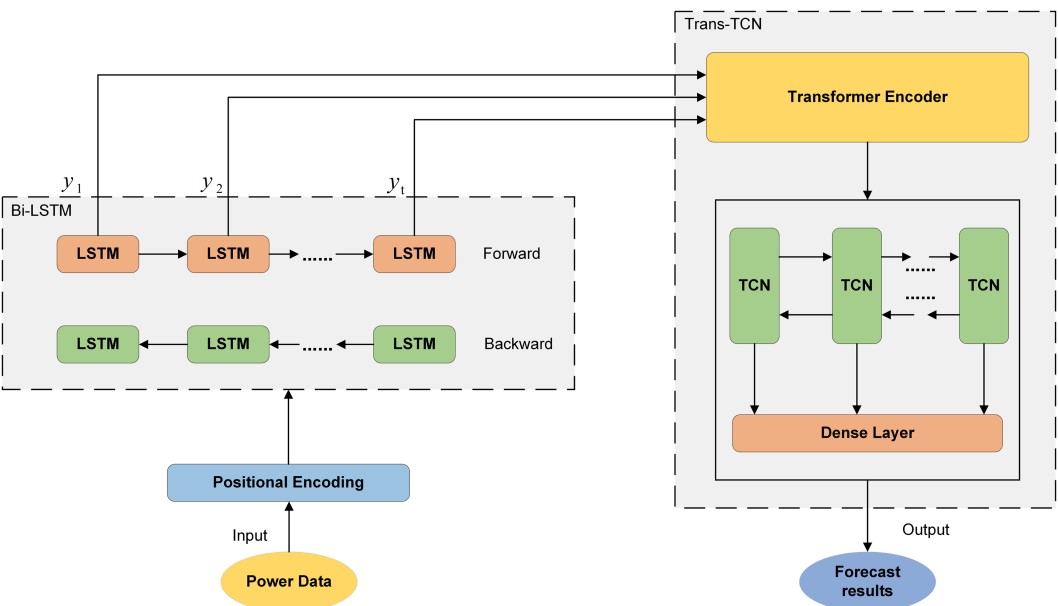

**Fig 2. BLTT-FT overall network structure.** The main structure consists of a position coding layer, a Bi-LSTM layer, a Transformer encoder layer, a TCN layer, and a fully connected layer.

them into a matrix $Q$, and packs the key vectors and value vectors into matrices $K$ and $V$. The output of the query vector is actually determined by the hidden vectors encoded in the previous layer, and the matrices $K$ and $V$ are assigned the same value as $Q$ in the self-concern. The result of the output calculation is the weighted value, which is derived from the compatibility operation between the query vector and the key vector. $f = \{f_i\}_{i=1}^{t}$ is the input to the multi-head self-attentive module. The key vectors, weight vectors, and query vectors are calculated as shown in Eq (1):

$$\begin{cases} K_j = fW_j^k \\ V_j = fW_j^k \\ Q_j = fW_j^q \end{cases} \tag{1}$$

where $W_j^k, W_j^k, W_j^q \in R^{d \times d_k}$ is the trainable projection matrix. The derived results are utilized and the Billet dot product attention calculation is performed as shown in Eq (2):

$$Attention(Q, K, V)_j = Softmax\left(\frac{Q_j K_j^T}{\sqrt{d_k}}\right)v_j \tag{2}$$

In order to jointly focus on the information from different representation subspaces at different locations, further optimization is required by using $H$ parallel attention computations, in the case of the multi-head attention mechanism computation process shown in Eq (3):

$$MultiHead(Q, K, V) = Concat\left(\{head_j\}_{j=1}^{H}\right)W^A \tag{3}$$

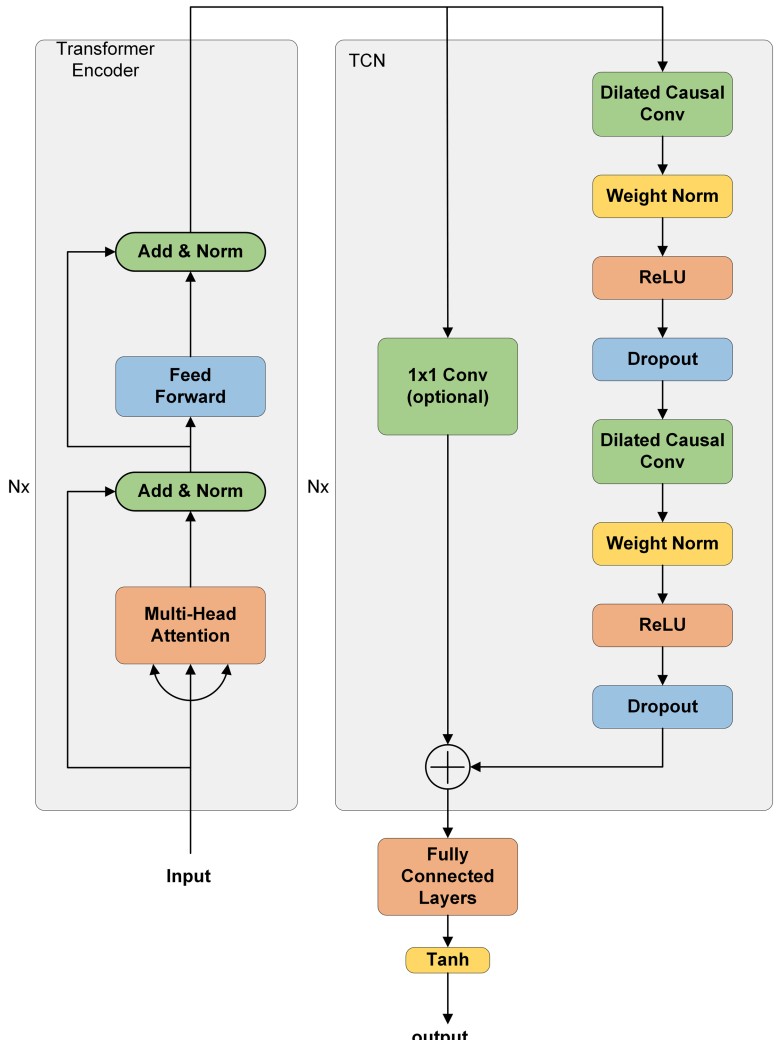

**Fig 3. Trans-TCN network structure.** The left half of the Trans-TCN network structure is the Transformer encoder, and the right half is the TCN and fully connected layer.

The functional realization of the coding layer also passes through the feedforward layer The feedforward network consists of two linear transformations connected in the middle by a ReLU activation function, which is applied separately and with the same status for each time step. The linear transformation formulas are shown in Eq (4):

$$y = W^T x + b \tag{4}$$

where $W^T$ is the weight matrix and $b$ is the bias vector.

(2) The TCN consists of multiple layers of residual blocks (RBs), and each RB is mainly composed of two layers: the dilated causal convolution (DCC), the weight initialization layer, and the ReLU. As shown in Fig 4A, the output of the Transformer encoder module is used as the input of the first layer of the TCN RB. Fig 4B shows the structure of DCC.

Among them, the weight initialization layer and the Droupt layer are mainly used to suppress the network noise and optimize the network training effect, and then in order to ensure that the input and output dimensions are the same, a 1x1

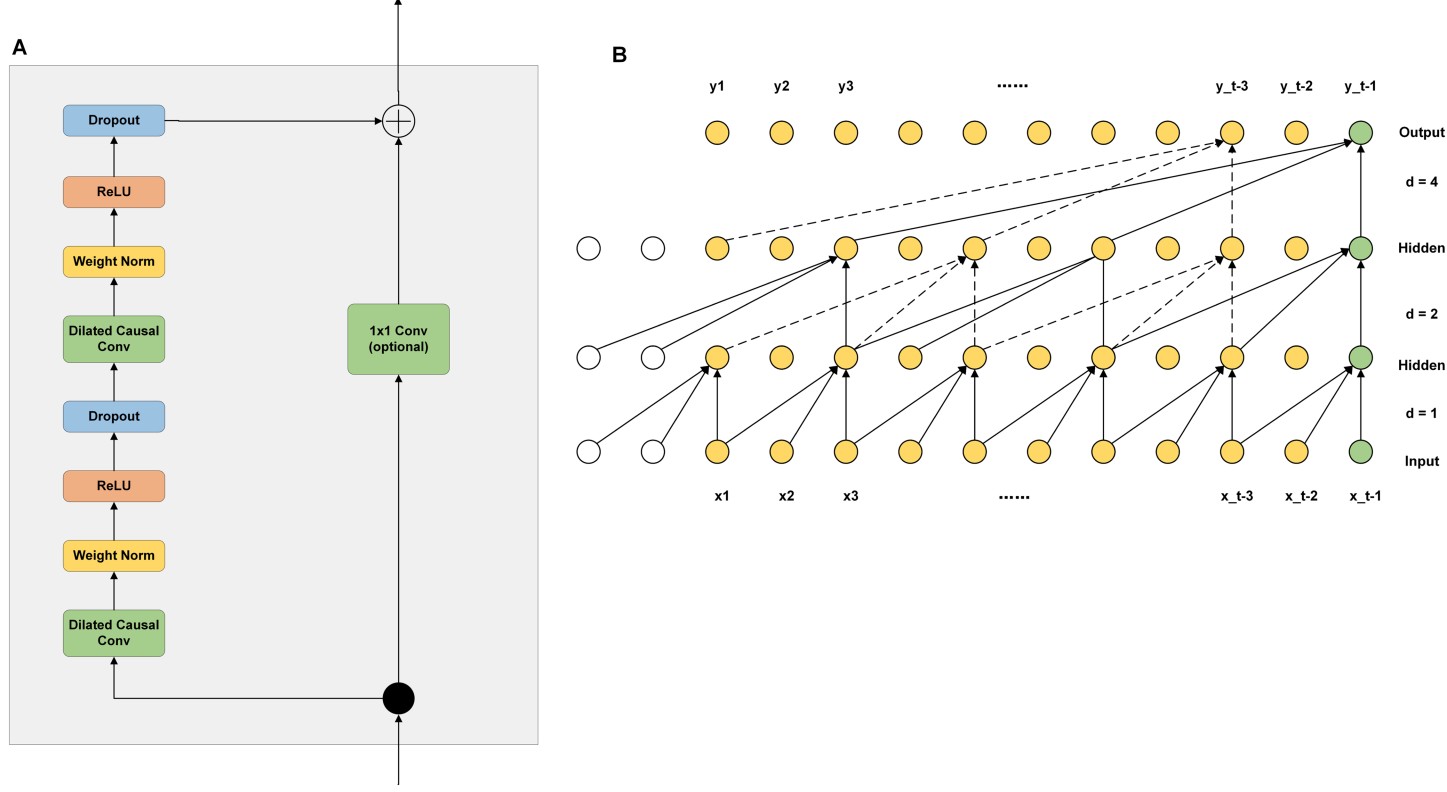

**Fig 4**. **TCN network structure.** A:Structure of TCN residual block. B:Structure of TCN expansion causal convolution.

convolution module is introduced, and finally the deep TCN can be constructed by continuously stacking the RBs. The combination of DCC and RB network structure can effectively improve the robustness and feature learning ability of TCN models.

Here is a brief introduction to DCC. Recurrent neural networks rely on historical data to process information and are unable to achieve massively parallel computation like CNNs, TCNs are an improved CNN that compensates for the short-comings of traditional CNNs in dealing with long-term data dependency by introducing DCC, which ensures that the model utilizes only historical data to make predictions and thus avoids interference from future data. The input sequence $\{x_0, x_1, ......, x_{t-1}\}$ undergoes DCC to obtain the output sequence $\{y_0, y_1, ......, y_{t-1}\}$, and $d$ is the number of expansions. Together, these elements improve the generalization ability of the model, accelerate the training process, and effectively prevent the overfitting problem, and the calculation of DCC is shown in Eq (5):

$$F(s) = \sum_{i=0}^{k-1} f(i) \cdot x_{s-d \cdot i} \tag{5}$$

where $\cdot$ is the convolutional computation, $d$ is the number of expansions, $s$ is the neuron, $k$ is the convolutional kernel, $f(i)$ is the $i$th data in the convolutional kernel,and $x_{s-d \cdot i}$ denotes the result of convolving the past data.

In summary, the parts of TCN that improve on the Transformer model are shown in the following:

1. Deleted "Input Embedding", which is a module for vectorization of language and text, is required for machine language translation and does not need to be vectorized for power.

2. Replace the transformer decoder with a TCN layer, a fully connected layer (FC-Linear), and an activation function Tanh (activation function).

3. The other inputs to the decoder are removed, leaving the output of the encoder as the only input to the decoder.

**Bi-LSTM network construction**

The BiLSTM neural network model is constituted by two independent LSTM networks, where the sequence information is fed into the network separately by means of forward and reverse order, making forward and backward information available to each LSTM unit.

(1) Recurrent Neural Networks (RNN) are deep neural networks used to predict time series, however, RNNs suffer from the problem of gradient vanishing, which is solved by the emergence of LSTM. The LSTM has three special gates, i.e., the forgetting gate $f_t$, the input gate $i_t$, and the output gate $o_t$, as well as the hidden unit $\tilde{C}_t$, which is used to regulate the information flow in the network. where $C_{t-1}$ is the output of the previously hidden cell, $C_t$ is the output of this hidden cell, $h_{t-1}$ is the output of the previous cell, $h_t$ is the output of this cell, $x_t$ is the input at this moment, $\sigma$ is the *sigmoid* function, and *tanh* is the hyperbolic tangent function.

Due to the large sample size of the dataset, the traditional LSTM, although performing well, leads to a large number of parameters that need to be learned for each gate, which results in a long-running time for the LSTM, so an improved LSTM network is proposed to speed up the data learning. The structure of the LSTM network is shown in Fig 5, and the structure of the improved LSTM network is shown in Fig 6.

The improved LSTM structure retains only 1 gate, which can effectively reduce the number of parameters while extracting the hidden timing information to ensure the accuracy of the model. The specific formula is shown in Eq (6):

$$
\begin{cases}
\tilde{C}_t = tanh(W_{x\tilde{C}} \cdot x_t + W_{h\tilde{C}} \cdot h_{t-1} + b_{\tilde{C}}) \\
tanh(x) = \frac{e^x - e^{-x}}{e^x + e^{-x}} \\
g_t = \sigma(W_{xg} \cdot x_t + W_{hg} \cdot h_{t-1} + P_{cg} \cdot C_{t-1} + b_g) \\
\sigma(x) = \frac{1}{1+e^{-x}} \\
C_t = (\tilde{C} + C_{t-1}) * g_t \\
h_t = g_t \cdot tanh(C_t)
\end{cases}
\tag{6}
$$

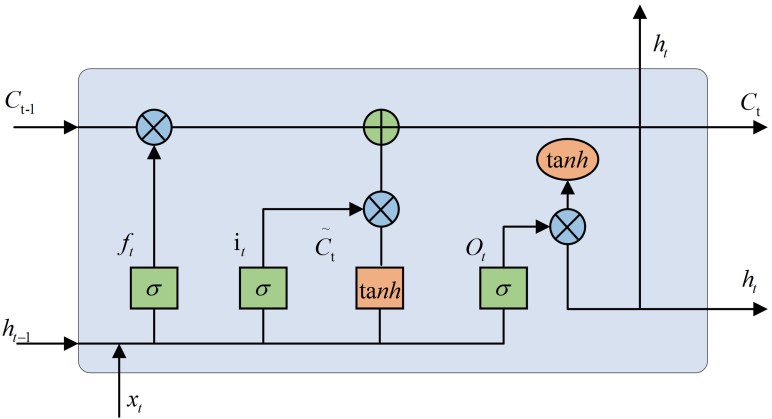

**Fig 5**. **LSTM network structure.** The LSTM has three special gates.

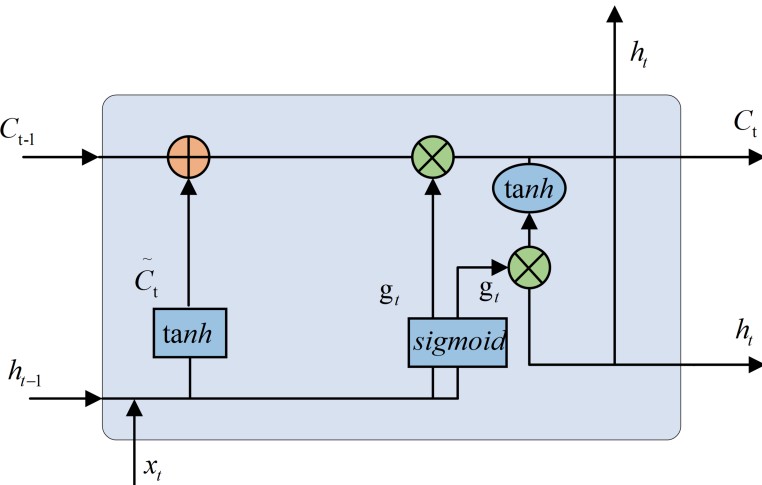

**Fig 6**. **Improved LSTM network structure.** The improved LSTM structure retains only 1 gate.

where: $X_t$ is the input; $H_{t-1}$ is the upper moment output; $W_{x\tilde{C}}$, $W_{h\tilde{C}}$, $W_{xg}$, $W_{hg}$ and $P_{cg}$ represent the corresponding weight matrices, respectively; $b_{\tilde{C}}$ and $b_g$ represent the bias vectors; and $g_t$ is the output gate.

(2) The BiLSTM model is constituted by the above improved LSTM, which is a network in which the output vectors (i.e., extracted feature vectors) of the forward LSTM and the backward LSTM are spliced to form a vector as the final output. The final output vector is calculated as shown in Eq (7):

$$y_t = [Z_t^f * Z_t^b] \tag{7}$$

where the output of the forward LSTM is $Z_t^f$ and the output of the backward LSTM is $Z_t^b$.

BiLSTM is modeled so that the feature data obtained at moment $t$ has information from the previous and subsequent moments, which results in better feature extraction efficiency and performance compared to a single LSTM structure. The model structure is shown in Fig 7, where $M_t$ is the weight matrix and $Z_t$ is the output vector. The BiLSTM model using the improved LSTM composition is called "Bi-LSTM" in this paper.

## Data preparation and processing

### Dataset construction

**Experimental sampling for fabric tearing performance testing.** This paper is based on the international standard "ISO 139372-2000 Textile fabrics-Tear properties-Part 2: Determination of tear strength of trouser specimens (single seam)" to carry out the test experiment of tear properties of trouser specimens. For each laboratory fabric samples should be cut 2 groups of specimens, a group of radial and a group of weft, specimens for the rectangular selection of samples, each group of samples should be 5 pieces of specimens, and every two pieces of specimens can't contain the same length or width of the direction of the yarn, should be avoided at the crease, the edge of the fabric and the fabric on the unrepresentative area, can't be taken in the distance from the edge of the fabric within 150mm sample from the lab samples to be cut from the sample is shown in Fig 8A for example. Trouser shaped specimens were (200±2) mm long and (50±1) mm wide. Each specimen should be cut from the middle of the width direction with a split of (100±1) mm long parallel to the length direction, and the end point of tearing was marked at (25±1) mm from the uncut end in the middle distance of the specimen as shown in Fig 8B. The denser yarns are the warp yarns and the less dense yarns are the weft

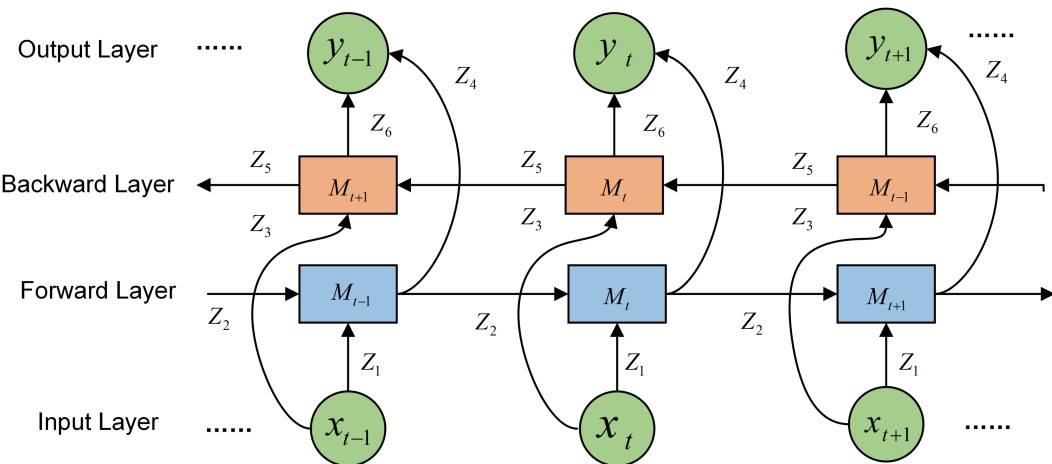

**Fig 7**. **Bi-LSTM network structure.** Bi-LSTM network structure composed of improved LSTMs.

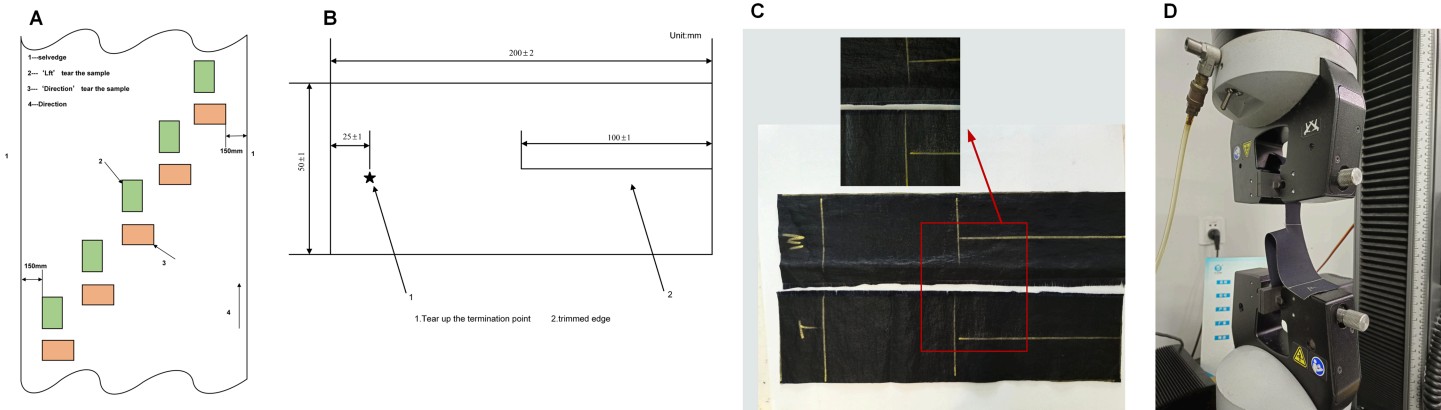

**Fig 8**. **Sampling diagram for fabric tearing test.** A: Example of a specimen cut from a laboratory sample. B: Standardized specimen charts. C: Detail of warp and weft yarns. D: Schematic diagram of tearing.

yarns, as shown in Fig 8C. The experimental principle of trouser specimen tearing performance testing is to clamp the two legs of the trouser specimen, so that the specimen notch line between the upper and lower fixtures into a straight line, to set the length of the spacing of the fabric tear tester to 100 mm, the tensile rate is set to 100 mm/min. open the instrument will be tensile force is applied in the direction of the notch, as shown in Fig 8D. All samples were cut from the middle of wrinkle-free and undamaged fabric bolts, avoiding areas within 150 mm of the fabric edges as well as defective regions such as yarn joints and stains. For samples taken from the same fabric, the warp/weft direction samples do not contain overlapping yarns to ensure the independence of each sample. After sampling, the samples were pre-treated in a constant temperature and humidity chamber (20±2 °c, 65±4% RH) for 24 hours to eliminate the impact of environmental factors on the physical properties of the fabrics.

**Data acquisition.** The fabric tearing performance test experiment adopts CRE isokinetic elongation tester to carry out the trouser specimen tearing performance test experiment. The CRE isokinetic elongation tester used was an Instron 5967 double-column bench-top tester.

For fabric tearing performance testing experiments, real-time monitoring of the power usage of the fabric tear tester, including current, voltage, and power parameters, was carried out, and one experiment was composed of five sets of radial and five sets of weft tearing experiments for each fabric. During the data perception of a single tearing experiment the CRE isokinetic elongation tester can obtain the time course of the four device states: The first is the equipment standby state, waiting for the experiment to be carried out, low-power mode; followed by the equipment sample change state, the installation of the specimen, low-power mode; followed by the equipment tensile state, subject to the different characteristics of different fabrics consume different power; and finally, the equipment reset state, the deformation of different fabrics is different, so that the device's pneumatic jig reset consumes a larger amount of power, specifically as shown in Table 1:

According to the working condition of the CRE isotropic elongation tester, the collection of electrical parameters needs to be carried out throughout the whole experimental process, especially the current power consumption in the electrical parameters is the largest when the equipment is working for stretching and it is mentioned in the international standard that for the isotropic elongation tester, if the strong force and elongation records are obtained through the data collection chip and software, then the frequency of data collection should be at least 8Hz. According to the Nyquist sampling theorem, to recover the original signal from the sampled signal without distortion, the sampling frequency should be greater than two times the highest frequency of the signal, so the design of power parameters should be collected at a frequency of not less than 16Hz, which can reflect the changes in the power load of the monitoring equipment in real-time. In the predictions of this paper, only the electric power is used as the prediction data because the electric power most clearly demonstrates the complete process of the entire fabric tearing performance testing experiment.

Environmental temperature and humidity will affect the physical properties of the fabric, in the high-temperature environment, the strength and toughness of the fabric will be reduced; in the humid environment, the softness and elasticity of the fabric will be reduced, and the fabric tearing performance testing requires a specific temperature and humidity conditions for testing. The international standard ISO 139-1973 Textiles-Standard atmospheres for conditioning and testing stipulates that the atmospheric temperature is 20.0°C with a tolerance of ±2.0% for temperature and 65.0% for relative humidity with a tolerance of ±4.0% for relative humidity. The electric power data of the experimental process of different fabric material samples were collected in a temperature and humidity range that ensures that it is a suitable temperature and humidity for doing the experiments, and the predictive analysis will be performed based on each individual experimental unit in order to improve the representativeness and generalization of the dataset.

### Preprocessing

**Missing value processing.** There are some missing values in the electric power parameters monitored during the experimental process of trouser fabric tearing performance testing (e.g., equipment failures, data acquisition problems, etc.), which leads to this problem may result in a decrease in prediction accuracy, and dealing with the missing values can make the data in the dataset more complete and improve the availability of the data and the reliability of the model.

Table 1. **Working status of CRE isokinetic elongation tester equipment.**

| Working State | Description |
|---|---|
| Standby state | The device is started and ready for experimentation, waiting for operational commands |
| Sample change state | Replacement of experimental samples, installation of new samples |
| Tensile state | Measurement of fabric tensile properties, including tensile strength, elongation, etc. |
| Reset state | Return to the initial position after completion of the experiment and prepare for the next round of experiments |

During the data perception process of a single tearing experiment, a CRE isotropic elongation tester can get the timing process of four device states: standby state, sample change state, tensile state, and reset state.

Due to the characteristics of serial data should not directly delete the missing values, this paper uses Last Observation Carried Forward (LOCF) to fill the later data with the data of the previous time.

**Smoothing.** The use of "mean padding of front and back values" for data smoothing is conducive to the elimination of isolated noise points in the time series data and the handling of minor missing value problems, smoothing of training data, elimination of burrs, and enhancement of the consistency and stability of the data.

**Data breach processing.** Data leakage is a phenomenon in which information from test sets or future data is used in the learning and training of a predictive model, causing the model to perform well in real-world applications but to lose predictive power on new data. To avoid this problem, the test set and training set need to be divided correctly. Before training the model, the original dataset is divided into a training set and a test set using a correlation random function, the model parameters are trained and adjusted by the training set, and the test set is used to evaluate the model generalization ability. The data sets were divided into the following proportions: 70%, 30%; 60%, 40%; 80%, 20%; 90%, 10% (training and test sets).

**Event-driven scrolling window segmentation.** In the timing prediction task of this paper, the timing data show an overall periodic trend, but the actual duration of each cycle is not fixed. In order to solve this problem, a rolling window concept that integrates the principle of discrete-time state machine and dynamic adjustment is innovatively proposed, focusing on the prediction of the multi-step state that occurs after a "change- tensile -reset" state. Specifically, a state window $W$ is defined, the length L of which aims to cover an expected cycle length and is used to contain all relevant information from the completion of the first "change- tensile -reset" state until the present moment. Yet while the overall trend is cyclical, the actual time span T of each cycle can fluctuate. Therefore, the update of the state window $W$ does not follow a fixed time sequence but is dynamically adjusted according to the occurrence of the "change- tensile -reset" state, which ensures that whenever such an operation occurs, the dataset $D_t$ contained in the window $W_n$ is strictly limited to the complete period from the end of the previous state to the beginning of the current state.

Next, the prediction model will be trained based on the data in the window $W_n$ to realize multi-step prediction, i.e., to predict the "change- tensile -reset" state after the current state. This process can be expressed as $M_n = Train(M, W_n)$, where the $Train$ function is responsible for performing the model training, and $M_n$ is the trained model, which has the ability to perform multi-step prediction, and at the same time can adapt to the change of cycle length.

In the prediction stage, the model $M_n$ predicts the state of "change- tensile -reset" at the moment $t_{n+x}$ based on the latest data $D(t_n)$, where x indicates the range of the prediction, e.g., 1 step, i.e., $p_{n+x} = predict_x(M_n, D(t_n))$, where the $predict_x$ function performs the prediction of the $x$ step. Although the cycle length $T$ may change, by dynamically adjusting the state window $W_n$, the model is able to capture and learn from these changes, thus improving the accuracy and robustness of the prediction, and the specific training process of the model is detailed in "Dataset construction".

Every time the "change- tensile -reset" state is completed, the state window scrolls to the next state point, and the model is updated, forming a closed loop of continuous iteration and optimization. This approach makes full use of the state information in the data to ensure that the model accurately captures and predicts the dynamics of the "change- tensile -reset" cycle, providing reliable multi-step predictions even in the face of irregular cycle lengths.

**Hierarchical 3-fold cross-validation design.** To eliminate the accidental bias caused by a single data division, supplementary hierarchical 3-fold cross-validation was conducted: with "fabric elasticity grade" as the hierarchical variable (the total dataset includes 150 types of high-elasticity fabrics, 170 types of medium-elasticity fabrics, and 160 types of low-elasticity fabrics, see Section 4.1 for data description), each fold contains 50 types of high-elasticity fabrics, 57 types of medium-elasticity fabrics, and 53 types of low-elasticity fabrics (160 types in total per fold). The division is based on fabric types without overlap, ensuring that the test set consists of "completely unseen new samples". For each fold, training (2 merged folds as the training set, 320 types of fabrics) and testing (1 fold as the test set, 160 types of fabrics) are performed independently, with the same parameters and independently initialized weights to avoid cross-fold interference. This design is complementary to the rolling window: the former verifies the model's adaptability to "brand-new fabrics", while the latter simulates the real-time scenario of "continuous addition of new data". Together, they ensure the reliability

of evaluation from two dimensions: sample diversity and temporal dynamics, and both strictly adhere to the principle that there is no information overlap between the training set and the test set.

## Experiment and parameter setting

### Data description

In the process of fabric tearing performance testing experiments, the equipment used is the Instron 5967 double-column bench-top tester, and in the whole experiment, the working state of the equipment can be divided into four states, which are: standby state, sample change state, tensile state, and reset state, as shown in Table 1 of "Research methods". Because the standby state has little correlation with the fabric tearing experiment, only the remaining three states are predicted in this paper.

The power visualization time series plot of the fabric tearing performance testing experiment is shown in Fig 9 below, where the shaded portion indicates the different states in the fabric tearing performance testing experiment. Fig 9A shows the changeover state, Fig 9B shows the stretching state, and Fig 9C shows the reset state.

On the time-series data in this paper, the principle of discrete-time state machines and the concept of dynamically adjusted rolling windows are integrated, focusing on predicting the multi-step state that occurs after a "change- tensile -reset" state. Set the "change- tensile -reset" state as a whole state, every time the "change- tensile -reset" state is completed, the state window will scroll to the next state point, and the model will be updated accordingly, forming a closed loop of continuous iteration and optimization.

Electrical power data were collected from the same fabric warp and weft tear performance testing experiments at a frequency of 20 Hz. Each prediction will focus on the results of the next "change- tensile -reset" operation. As shown in Fig 10, $x_1$ represents a "change- tensile -reset" operation, which generates $\{x_1, x_1, ......, x_n\}$ sequence of temporal features as input to the model.

Based on the power of the experimental process of tearing fabrics with three different categories of high, medium, and low elasticity, the data sets were divided into three categories, as shown in Table 2 below:

### Model training and parameter setting

In this study, the experimental system is running on Windows 11 operating system with Intel(R) Core(TM) i5-8250U CPU @ 1.60GHz processor, 16.0GB of system memory, Python version 3.9 installed in the experimental environment, and Pytorch version 1.13.1.

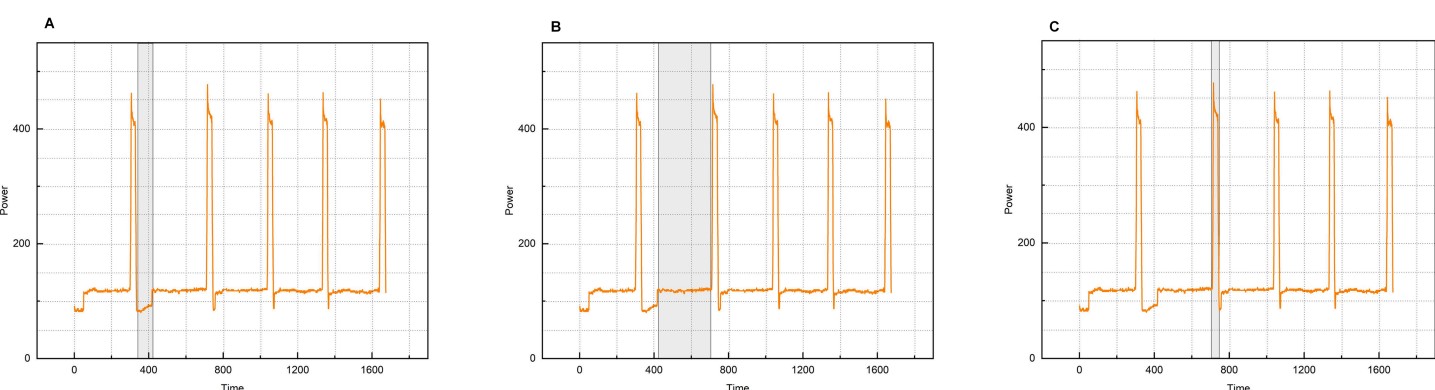

**Fig 9**. **Sequence diagram of power visualization.** A:Sample change state. B:Tensile state. C:Reset state.

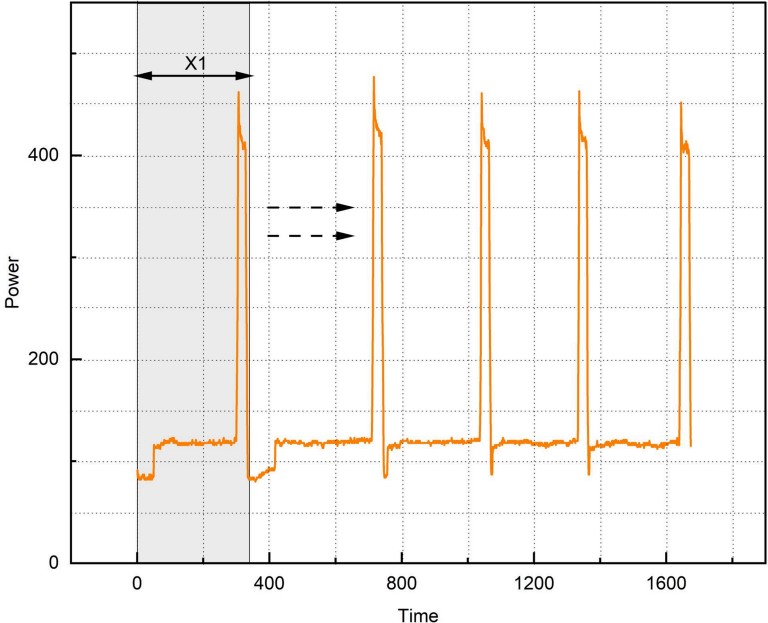

**Fig 10**. **Scrolling time window.** $x_1$ represents a "change- tensile -reset" operation, which generates $\{x_1, x_1, ......, x_n\}$ sequence of temporal features as input to the model.

**Table 2**. **Three types of data sets.**

| Dataset | Fabric Performance | Fiber Type | Sample Size |
|---------|--------------------|-----------|-------------|
| 1 | High elasticity | Nylon, spandex, lycra, recycled cellulose, elastic polyester ...... | 150 types, total 1500 pieces |
| 2 | Middle elasticity | Wool viscose fiber blend, nylon spandex blend ...... | 170 types, total 1700 pieces |
| 3 | Low elasticity | Polyester, cotton ...... | 160 types, total 1600 pieces |

The number of samples refers to the total number of fabric types and their corresponding specimens. In accordance with international standards, for each type of fabric, 5 specimens are taken along the warp direction and 5 along the weft direction (with dimensions of 200 ± 2mm × 50 ± 1mm), resulting in a total of 10 specimens per fabric type. Therefore, Dataset 1 (150 fabric types) contains a total of 150 × 10 = 1500 specimens, Dataset 2 (170 fabric types) contains 170 × 10 = 1700 specimens, and Dataset 3 (160 fabric types) contains 160 × 10 = 1600 specimens.

The BLTT-FT model proposed in this paper is first trained offline. The datasets used are datasets 1, 2 and 3 described in Table 2, taking dataset 1 as an example: the whole dataset collects the electric power acquisition signals of the experimental process of testing the tearing performance of fabrics with fabric properties of high elasticity, in chronological order. There are five warp direction specimens and five weft direction specimens for one fabric, in order to increase the generalization of the experiments, there are 150, 170 and 160 types of fabrics with high, medium and low elasticity, respectively, and a total of ten experiments are done for each type of doing warp and weft, which basically includes most of the fabric fabrics and ensures the diversity of the fabric types to achieve wider applicability. The long data segment is composed of the electric power signals collected during the experimental process of 10 specimens, the medium data segment is composed of the electric power signals collected during the experimental process of the longitudinal and latitudinal specimens, respectively, and the short data segment is composed of the electric power signals collected during the experimental process of the first group of radial and the first group of latitudinal specimens. From the above, it can be seen that each dataset has been categorized into long, medium, and short data segments, and each dataset is fed in parallel to

the BLTT-FT model for offline training. In the online application, when the first set of electric power data of the experimental process is obtained, every time the "change- tensile -reset" operation is completed, the state window will scroll to the next state point, and the model will be updated accordingly, and the incremental learning will be repeated, updating the database to form a continuous iteration and optimization of the closed loop. It is used to predict the next electric power data. The specific flow is shown in Fig 11.

The parameters for experimental training are as follows: epoch = 100, batch size is set to 32, the initial learning rate of Adam is set to 0.00001, the ReLU activation function is adopted, the loss function is the MSE function, and predictions are made for the ranges of 1, 4, and 9 steps respectively. The cross-validation parameters are consistent with the above, with only the data division method adjusted: the training set and test set of each fold strictly follow the hierarchical 3-fold rules, and the model parameters of each fold are initialized independently to avoid cross-fold information leakage. The parameters of the proposed model are shown in Table 3.

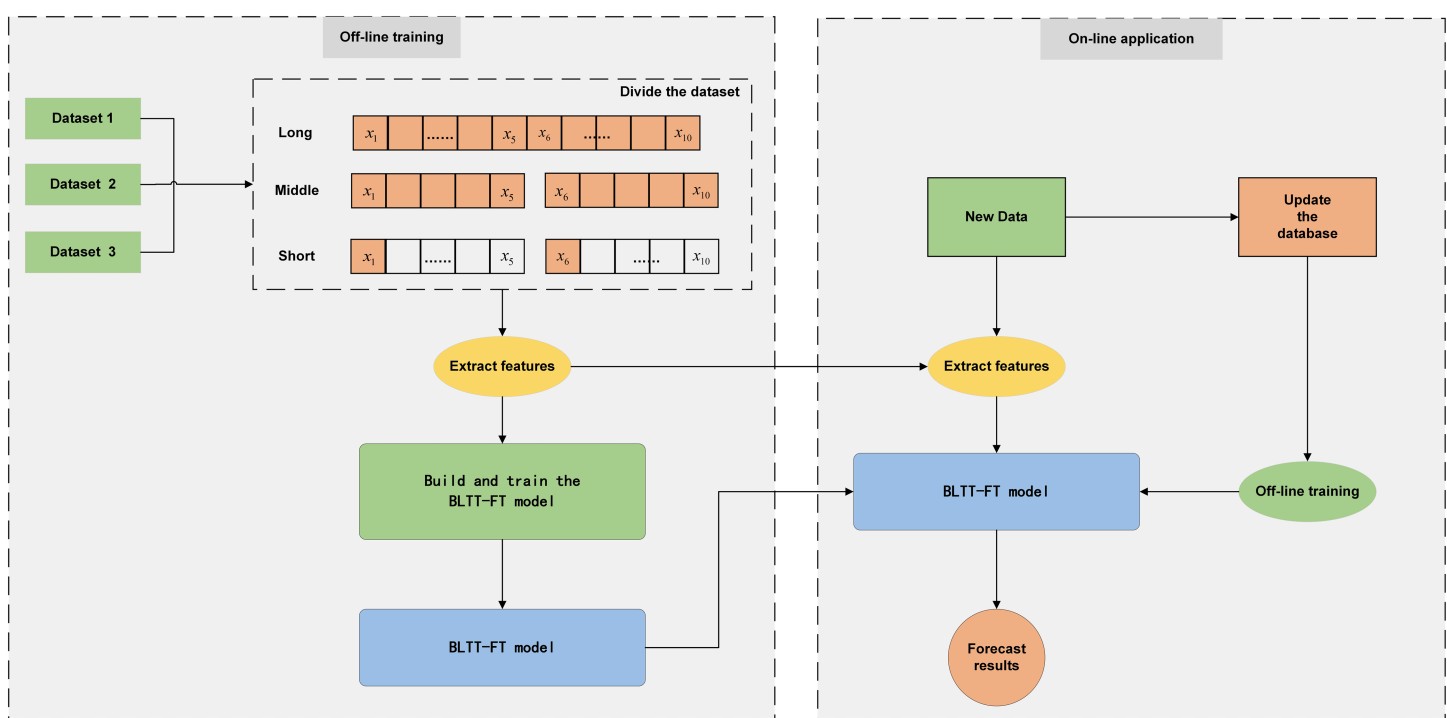

**Fig 11**. **Offline training-online application flowchart.** Flowchart of BLTT-FT model for offline training and online application.

**Table 3**. **Parameter settings for the BLTT-FT model.**

| Parameter Settings | Parameter Settings |
|---|---|
| Batch size = 32 | Sequence length of training data = [1,4,9] |
| Hidden size of BiLSTM = 64 | Num layers = 1 |
| Dropout = 0.5 | Number of transformer encoder head = 8 |
| Number of transformer encoder layer = 6 | TCN layer kernel size = 1 |
| Number of TCN layer neurons = 32 | Number of TCN hidden layer = 4 |
| Kernel size of TCN layer = 7 | Activation function of TCN layer = ReLU |

The training parameters of the BLTT-FT model are shown in detail.

In the offline training phase, the BLTT-FT model completes 100 epoch training based on Windows 10 system (Intel i5-8250U CPU, 16GB RAM) with a total time of 8 hours, and the test set loss curves are highly fitted to the training set, as shown in Fig 12, indicating that the model is not overfitted. The loss curve of BLTT-FT not only decreases considerably, but also has a shorter fitting time, and the loss curve of the test set is closer to the training set, with excellent training results.

In online application, whenever the "change - tensile - reset" operation is completed, the model realizes incremental learning by scrolling through the state window, and a single update takes about 200 ms, and the prediction error after the update is reduced by 3.2% on average compared with that before the update, which demonstrates the fast adaptability to real-time data. The performance data of the two modes are shown in Table 4.

## Experimental results and discussion

### Indicators for model evaluation

Mean Absolute Error (MAE), Mean Square Error (MSE), Root Mean Square Error (RMSE), and Mean Absolute Percentage Error (MAPE) were used as the evaluation criteria for the method, and these error evaluation metrics were calculated

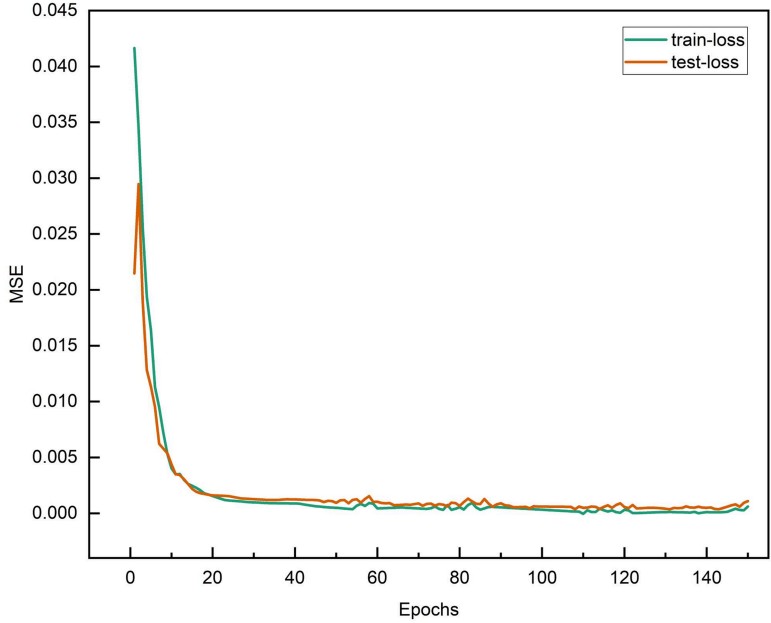

**Fig 12. BLTT-FT's loss curve.** The loss function curve shows the trend of the loss value of the model during the training process, which helps to determine whether the model is overfitting or underfitting in order to adjust the model structure and training strategy.

**Table 4**. **Performance data in two modes.**

| Learning Model | Key Indicators | Numerical Value |
|---|---|---|
| Offline Training | Total training time | 8h |
| Online Application | Single incremental learning time | 200ms |
| | Error reduction | 3.2% |

The table shows offline training time and online metrics like incremental learning duration and error reduction.

as shown in Eq (8):

$$
\begin{cases}
MAE = \frac{1}{n} \sum_{t=1}^{n} |\hat{y}_t - y_t| \\
MSE = \frac{1}{n} \sum_{t=1}^{n} (\hat{y}_t - y_t)^2 \\
RMSE = \sqrt{\frac{\sum_{t=1}^{n} |\hat{y}_t - y_t|^2}{n}} \\
MAPE = \frac{100\%}{n} \sum_{t=1}^{n} \left| \frac{\hat{y}_t - y_t}{y_t} \right| \\
R^2 = 1 - \frac{\sum_{t=1}^{n} (\hat{y}_t - y_t)^2}{\sum_{t=1}^{n} (\bar{y}_t - y_t)^2}
\end{cases}
\tag{8}
$$

In the above equation, $\hat{y}_t$ is the predicted value, $y_t$ is the actual value, $\bar{y}_t$ is the mean value, and $n$ is the number of samples. Generally speaking, the smaller the values of MAE, MSE, RMSE, MAPE, and the closer $R^2$ is to 1, the smaller the error between predicted and actual power is, indicating the better prediction performance of the model.

**Comparative experiments**

In order to verify the effectiveness of the BLTT-FT model in monitoring the power changes during the fabric tearing performance monitoring experiments, five common prediction models were selected for comparative analysis, which include LSTM model, BiLSTM model, Transformer model, TCN model, and Transformer-LSTM model for comparison of prediction performance. The parameters of each comparison model are shown in Table 5. And the power of the experimental process of tearing performance of fabrics with three different categories of high elasticity, medium elasticity, and low elasticity was collected within the range of atmospheric temperature of 20.0-22.0°C and relative humidity of 61.0-69.0% and was divided into three types of datasets, in which the electrical power data of the experimental process was predicted for the warp and weft fabrics of the same category of fabrics. Considering that the sample data is relatively limited, the ratio of 9:1 is used to divide the training set and test set to ensure that the model can be adequately trained.

The sequence data of each data segment in the dataset were input into the model for training, and the loss function descent curves of the training and test sets during the training of the five models were obtained as shown in Fig 13.

As seen in Fig 13, in LSTM, BiLSTM, Transformer, and TCN models, the training set loss curves still tend to decrease slightly in the late iterations, while the loss curves of the test set slightly increasing, which may lead to overfitting phenomenon if the number of iterations increases. However, the Transformer-LSTM model's loss curve does not have this phenomenon, and the validation set loss function decreases more than the LSTM, BiLSTM, Transformer, and TCN models, but the fitting time is relatively long. From Fig 12, it can be seen that the loss curve of BLTT-FT not only decreases more, but also has a shorter fitting time, and the test set loss curve is closer to the training set, which is a better training result.

As can be seen in Tables 6, 7, 8, in datasets 1, 2, and 3: the average RMSE for 1-step prediction is 0.0691, the average MAE is 0.0467, the average MAPE is as low as 2.31%, and the average $R^2$ was as high as 0.9865; the average

**Table 5. Different model parameter settings.**

| Model | Parameter Settings |
|---|---|
| LSTM | Hidden size=64; Num layers=2 |
| BiLSTM | Hidden size=64; Num layers=1 |
| Transformer | Num layers=1; Num heads=8; Dropout=0.5 |
| TCN | Kernel size=3; N levels=2; N hid=32; Dilation factor=[1,2] |
| Transformer-LSTM | Encoder layers= [1,8]; Dropout=0.5; Hidden size=64; Num layers=2 |

The training parameters of the other models are shown in detail.

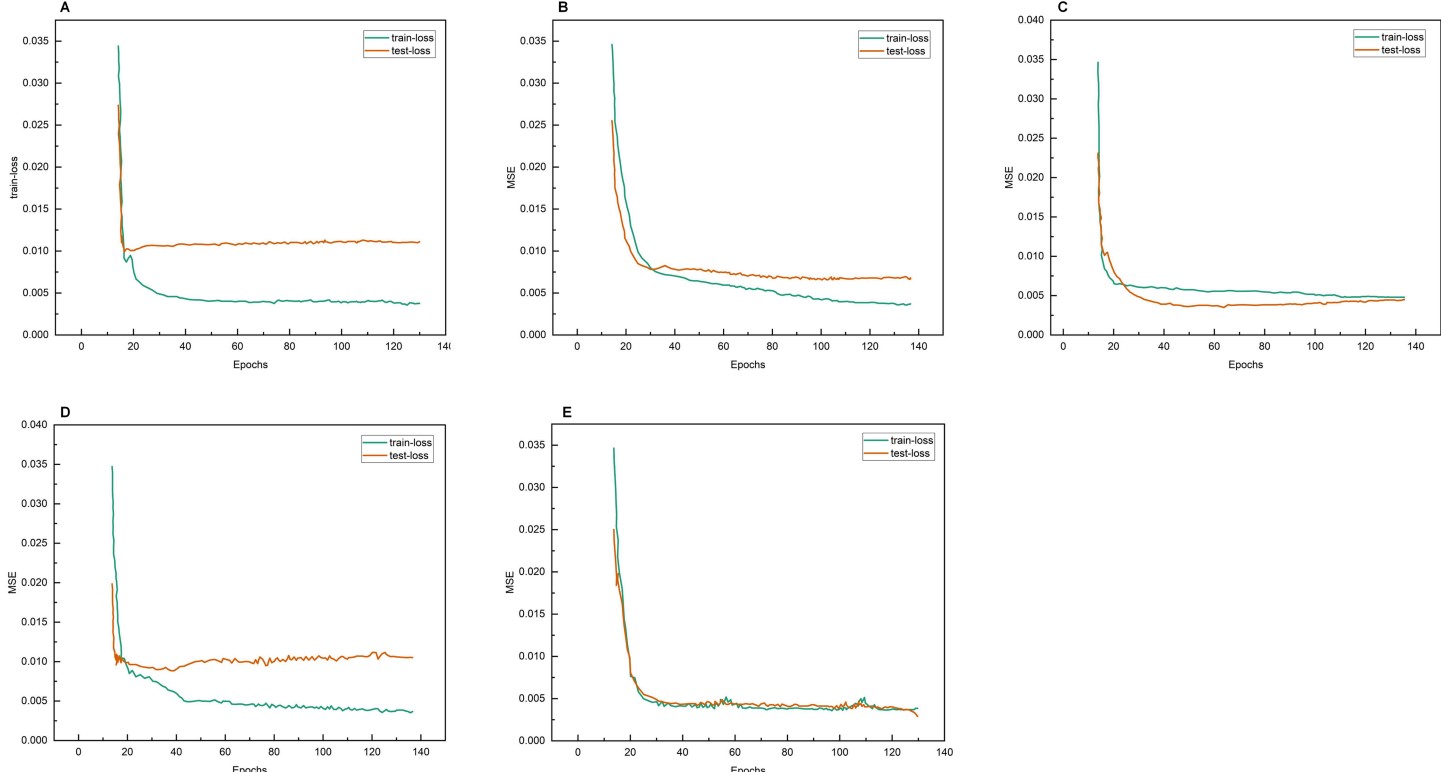

**Fig 13**. **Changes in loss curves during the fitting of the 5 models.** A: LSTM's loss curve. B: BiLSTM's loss curve. C: Transformer's loss curve. D: TCN's loss curve. E: Transformer-LSTM's loss curve.

RMSE for the 4-step prediction is 0.0911, the average MAE is 0.0632, the average MAPE is as low as 3.12%, and the average $R^2$ is as high as 0.9600; the average RMSE for the 9-step prediction is 0.1040, the average MAE is 0.0729, the average MAPE is as low as 3.74%, and the average $R^2$ is as high as 0.9252. In these three datasets, the proposed prediction method achieved high prediction accuracy for the prediction of electric power parameters during the experimental process of fabric tearing performance testing for different elasticity levels. As the prediction time step increases, the accuracy of the BLTT-FT based method will decrease but still maintain a high prediction accuracy. Moreover, the proposed method uses only the previous 1-step electric power data to achieve high-precision prediction for the next 9 steps. These results show that the BLTT-FT-based method has excellent performance and strong generalization ability in the prediction of electric power parameters during fabric tearing experiments with different elasticity levels.

The multi-step prediction errors of BLTT-FT in in Tables 6, 7, 8 above are visualized as shown in Fig 14A, 14B, 14C.The smaller the values of MAE, MSE, RMSE, MAPE, and the closer the $R^2$ is to 1, the smaller the error between the predicted power and the actual power is, which indicates that the prediction performance of the model is better. It can be seen that the BLTT-FT model in this paper has excellent prediction effect in the three datasets and stable prediction effect in different datasets.

In datasets 1, 2, and 3, the comparison of the prediction curves of the BLTT-FT based method with the actual curves are shown in Figs 15, 16, 17, demonstrating the comparison of the prediction results of different models with different prediction ranges. Figs 15A, 15B, 16A, 16B, 17A, 17B show the prediction of longitude and latitude fabrics at 1 step, Figs 15C, 15D, 16C, 16D, 17C, 17D show the prediction of longitude and latitude fabrics at 4 steps, and Figs 15E, 16E, 17E show the prediction of a whole set of fabrics at 9 steps. In these datasets, from 1 step prediction to 9 steps

**Table 6**. Evaluation metrics for dataset 1 in different models with multi-step prediction.

| Dataset | Model | Evaluation Metrics | Forecast Range | | |
|---|---|---|---|---|---|
| | | | 1 step | 4 steps | 9 steps |
| 1 | LSTM | MAE/mm | 0.1144 | 0.1612 | 0.1796 |
| | | RMSE/mm | 0.1471 | 0.2141 | 0.2431 |
| | | MAPE/% | 0.8854 | 0.8362 | 0.7863 |
| | | $R^2$ | 0.8854 | 0.8362 | 0.7863 |
| | BiLSTM | MAE/mm | 0.1102 | 0.1588 | 0.1778 |
| | | RMSE/mm | 0.1446 | 0.2141 | 0.2417 |
| | | MAPE/% | 5.78 | 8.53 | 9.72 |
| | | $R^2$ | 0.8943 | 0.8374 | 0.7890 |
| | Transformer | MAE/mm | 0.1196 | 0.1632 | 0.1767 |
| | | RMSE/mm | 0.1540 | 0.2192 | 0.2414 |
| | | MAPE/% | 6.37 | 8.82 | 9.80 |
| | | $R^2$ | 0.8424 | 0.8396 | 0.7887 |
| | TCN | MAE/mm | 0.1134 | 0.1629 | 0.1782 |
| | | RMSE/mm | 0.1492 | 0.2164 | 0.2421 |
| | | MAPE/% | 6.05 | 8.84 | 9.74 |
| | | $R^2$ | 0.8890 | 0.8391 | 0.7893 |
| | Transformer-LSTM | MAE/mm | 0.0984 | 0.1063 | 0.1112 |
| | | RMSE/mm | 0.1261 | 0.1361 | 0.1435 |
| | | MAPE/% | 5.31 | 5.75 | 6.05 |
| | | $R^2$ | 0.9013 | 0.8949 | 0.8890 |
| | **BLTT-FT** | MAE/mm | **0.0489** | **0.0741** | **0.0762** |
| | | RMSE/mm | **0.0625** | **0.0960** | **0.1001** |
| | | MAPE/% | **2.58** | **3.93** | **4.12** |
| | | $R^2$ | **0.9823** | **0.9437** | **0.9211** |

The prediction error values for dataset 1 at different prediction ranges for different models are shown in detail.

prediction, the error between the predicted curve and the actual curve increases as the prediction range increases, because the larger the prediction range, the greater the amount of missing information and the lower the prediction accuracy. However, the proposed BLTT-FT based method still has a high fit between the actual and predicted curves in steps 1, 4, and 9.

Based on the above analysis, the proposed model shows excellent performance in terms of prediction accuracy and stability, especially in the long-term prediction task.

As shown by the prediction graphs and evaluation indexes with prediction ranges of 1, 4, and 9 steps, the BLTT-FT prediction model proposed in this paper is excellent, and it can realize the conception of doing only the first group of tearing experiments in fabric tearing experiments to predict the fabric tearing experiments of the following groups, simplify the experimental process, and improve the efficiency and accuracy of the experiments.

## Significance test analysis

In the performance assessment of prediction models, it is difficult to determine whether the differences between different datasets are statistically significant by only comparing the numerical magnitude of the prediction error metrics. In order to scientifically assess the performance of the model on different datasets, this study analyzes the significance test of MAE, RMSE and MAPE metrics for three datasets under one-step, four-step and nine-step prediction. The prediction error metrics of different datasets under each prediction step are visualized by box-and-line diagrams to visually present the data distribution characteristics. Figs 18, 19 and 20 show the distributions of MAE, RMSE and MAPE for different datasets under each prediction step, respectively.

**Table 7**. Evaluation metrics for dataset 2 in different models with multi-step prediction.

| Dataset | Model | Evaluation Metrics | Forecast Range | | |
|---|---|---|---|---|---|
| | | | 1 step | 4 steps | 9 steps |
| 2 | LSTM | MAE/mm | 0.0968 | 0.1447 | 0.1727 |
| | | RMSE/mm | 0.1789 | 0.2611 | 0.3108 |
| | | MAPE/% | 4.25 | 7.27 | 8.92 |
| | | $R^2$ | 0.9102 | 0.8194 | 0.8092 |
| | BiLSTM | MAE/mm | 0.0887 | 0.1542 | 0.1827 |
| | | RMSE/mm | 0.1729 | 0.2626 | 0.3115 |
| | | MAPE/% | 3.96 | 7.40 | 9.00 |
| | | $R^2$ | 0.9432 | 0.8190 | 0.8090 |
| | Transformer | MAE/mm | 0.0898 | 0.1489 | 0.1749 |
| | | RMSE/mm | 0.1753 | 0.2671 | 0.3162 |
| | | MAPE/% | 4.13 | 7.47 | 9.10 |
| | | $R^2$ | 0.9211 | 0.8187 | 0.8087 |
| | TCN | MAE/mm | 0.1022 | 0.1475 | 0.1756 |
| | | RMSE/mm | 0.1845 | 0.2623 | 0.3117 |
| | | MAPE/% | 4.75 | 7.42 | 9.07 |
| | | $R^2$ | 0.9002 | 0.8188 | 0.8088 |
| | Transformer-LSTM | MAE/mm | 0.0862 | 0.0904 | 0.0992 |
| | | RMSE/mm | 0.1512 | 0.1568 | 0.1691 |
| | | MAPE/% | 4.04 | 4.28 | 4.72 |
| | | $R^2$ | 0.9219 | 0.9121 | 0.9011 |
| | **BLTT-FT** | MAE/mm | **0.0439** | **0.0521** | **0.0636** |
| | | RMSE/mm | **0.0865** | **0.0931** | **0.1079** |
| | | MAPE/% | **1.99** | **2.26** | **2.85** |
| | | $R^2$ | **0.9921** | **0.9854** | **0.9420** |

The prediction error values for dataset 2 at different prediction ranges for different models are shown in detail.

From Fig 18, it can be seen that the median MAE of dataset 2 is relatively low in one-step prediction; in four-step prediction, dataset 2 still shows a low median MAE; and in nine-step prediction, the median MAE of dataset 2 is also at a low level. This indicates that the prediction error of dataset 2 is relatively small and the prediction result is more accurate in terms of MAE. At the same time, the degree of data dispersion (box height and whisker length) varies among datasets at different prediction steps, reflecting the differences in the stability of prediction errors. For example, dataset 3 has a larger box height at nine prediction steps, which indicates that it has a high degree of data dispersion and poorer stability of prediction error.

Observing Fig 19, the median RMSE of dataset 3 is the lowest in one-step prediction, and the median RMSE of dataset 3 is relatively low in four-step prediction. In nine-step prediction, the median RMSEs of dataset 1, dataset 2, and dataset 3 are relatively close to each other, but there are high outliers in dataset 2 and low outliers in dataset 3. This indicates that dataset 3 has some advantages in RMSE indicators, especially in one-step prediction, and the prediction error is relatively small, but in nine-step prediction, the performance difference of each dataset is not obvious and there are outliers.

As shown in Fig 20, the median MAPE of dataset 2 is mostly the lowest under each prediction step. For one-step prediction, the median MAPE of dataset 2 is significantly lower than that of other datasets; for four- and nine-step prediction, dataset 2 also maintains a low median MAPE. This indicates that the prediction error of dataset 2 is relatively small under the MAPE indicator, and the model's prediction accuracy of the data is better in this indicator dimension.

One-way analysis of variance (ANOVA) was used to test the significance of the MAE, RMSE and MAPE indicators for different datasets at each prediction step, and the test results are summarized in Table 9.

**Table 8**. Evaluation metrics for dataset 3 in different models with multi-step prediction.

| Dataset | Model | Evaluation Metrics | Forecast Range | | |
|---------|-------|--------------------|--------|--------|--------|
| | | | 1 step | 4 steps | 9 steps |
| 3 | LSTM | MAE/mm | 0.1160 | 0.1867 | 0.2192 |
| | | RMSE/mm | 0.1729 | 0.2728 | 0.3226 |
| | | MAPE/% | 6.04 | 9.98 | 12.4 |
| | | $R^2$ | 0.8890 | 0.7879 | 0.7459 |
| | BiLSTM | MAE/mm | 0.1217 | 0.1794 | 0.2121 |
| | | RMSE/mm | 0.1793 | 0.2728 | 0.3226 |
| | | MAPE/% | 5.38 | 8.29 | 11.48 |
| | | $R^2$ | 0.9011 | 0.8372 | 0.7466 |
| | Transformer | MAE/mm | 0.1104 | 0.1736 | 0.2206 |
| | | RMSE/mm | 0.1695 | 0.2627 | 0.3268 |
| | | MAPE/% | 5.27 | 8.92 | 12.47 |
| | | $R^2$ | 0.9013 | 0.8352 | 0.7457 |
| | TCN | MAE/mm | 0.1395 | 0.2115 | 0.2546 |
| | | RMSE/mm | 0.1983 | 0.2941 | 0.3523 |
| | | MAPE/% | 8.87 | 12.99 | 15.60 |
| | | $R^2$ | 0.8350 | 0.7440 | 0.7267 |
| | Transformer-LSTM | MAE/mm | 0.0923 | 0.0977 | 0.0998 |
| | | RMSE/mm | 0.1299 | 0.1366 | 0.1397 |
| | | MAPE/% | 4.49 | 4.76 | 4.91 |
| | | $R^2$ | 0.9006 | 0.9001 | 0.8988 |
| | **BLTT-FT** | MAE/mm | **0.0472** | **0.0635** | **0.0791** |
| | | RMSE/mm | **0.0583** | **0.0842** | **0.1040** |
| | | MAPE/% | **2.34** | **3.17** | **4.26** |
| | | $R^2$ | **0.9851** | **0.9510** | **0.9124** |

The prediction error values for dataset 3 at different prediction ranges for different models are shown in detail.

The ANOVA results show that there are significant differences (p-value less than 0.05) between different datasets at each prediction step (1, 4, and 9 steps) under the three indicators of MAE, RMSE, and MAPE. This indicates that statistically speaking, the prediction errors of different datasets at each prediction step are not caused by random factors, but there are real and significant differences, and the F-value reflects the ratio of the between-group variance to the within-group variance, the larger the F-value is, the larger the degree of difference between the different datasets is compared with the degree of difference within the datasets, which further supports the conclusion that the datasets have significant differences in the corresponding indexes and the prediction steps. The

A one-way ANOVA was conducted to test the MAE, RMSE and MAPE indicators of different samples under each prediction step, and the results showed that there were extremely significant differences among different samples under all indicators and prediction steps.

The significance of MAE, RMSE and MAPE indicators of the three datasets under 1, 4 and 9 steps of prediction is tested by ANOVA, and the results show that there are extremely significant differences in all indicators and prediction steps, indicating that the differences between different datasets are not random. From the box plot visualization results, the corresponding dataset of this algorithm has outstanding performance in the error indicators, such as the MAE value is significantly lower than other datasets under most prediction steps. Combined with the significant results of ANOVA, it strongly proves that the present algorithm is significantly different from other methods in terms of prediction accuracy, and more superior in terms of statistical significance. At the same time, the information of box location and degree of dispersion of the corresponding dataset boxes of this algorithm in the box-line diagram further aids in arguing its superiority in prediction accuracy and stability from the perspective of intuitive error distribution, which provides a solid data and statistical basis for the validity and reliability of this algorithm in practical applications.

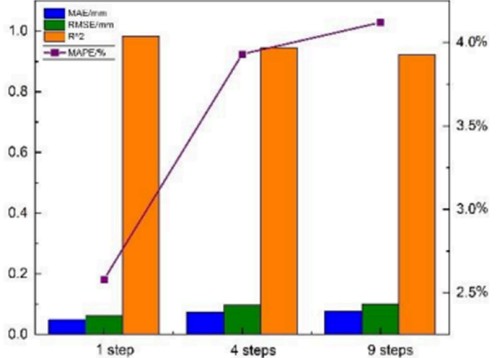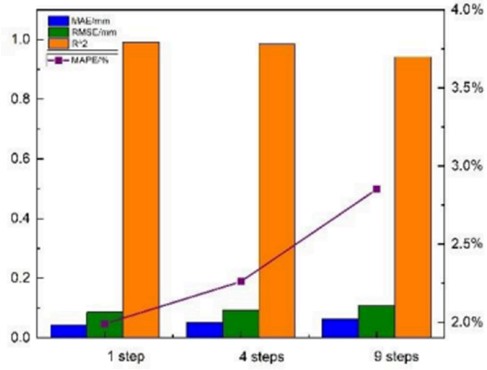

A Multi-step error visualization plot for dataset 1 B Multi-step error visualization plot for dataset 2

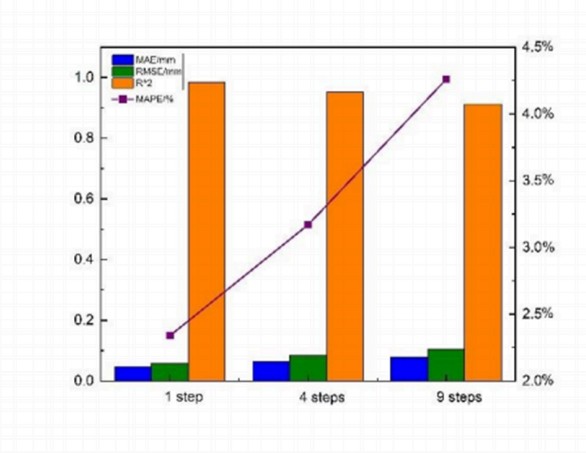

C Multi-step error visualization plot for dataset 3

**Fig 14. Multi-step prediction error visualization plots for datasets (1–3).** A: Multi-step error visualization plot for dataset 1. B: Multi-step error visualization plot for dataset 2. C: Multi-step error visualization plot for dataset 3.

## Ablation experiment

In order to comprehensively verify the performance contribution of each module in the BLTT-FT model, this paper investigates it through a hierarchical progressive ablation experiment design. First, focusing on the necessity of the core modules, Bi-LSTM, Transformer and TCN are removed for comparison respectively: by comparing BLTT-FT with Trans-TCN, and Bi-LSTM-TCN with TCN, we verify the key role of the Bi-LSTM module in capturing bi-directional sequence dependency; Highlighting the global attention advantage of the Transformer module in long-distance dependency modeling by comparing BLTT-FT with Bi-LSTM-TCN and Trans-TCN with TCN; Comparing BLTT-FT with Bi-LSTM-Transformer and Bi-LSTM-TCN with Bi-LSTM, the effectiveness of the TCN layer for local temporal feature extraction is clarified. Secondly, to verify the synergistic effect of multi-module combination, BLTT-FT is compared with single-module Transformer, Bi-LSTM and TCN, respectively, to analyze the overall improvement effect of multi-module fusion in enhancing the prediction performance. The model naming is shown in Table 10, and the experimental results are shown in Tables 11, 12 and 13.

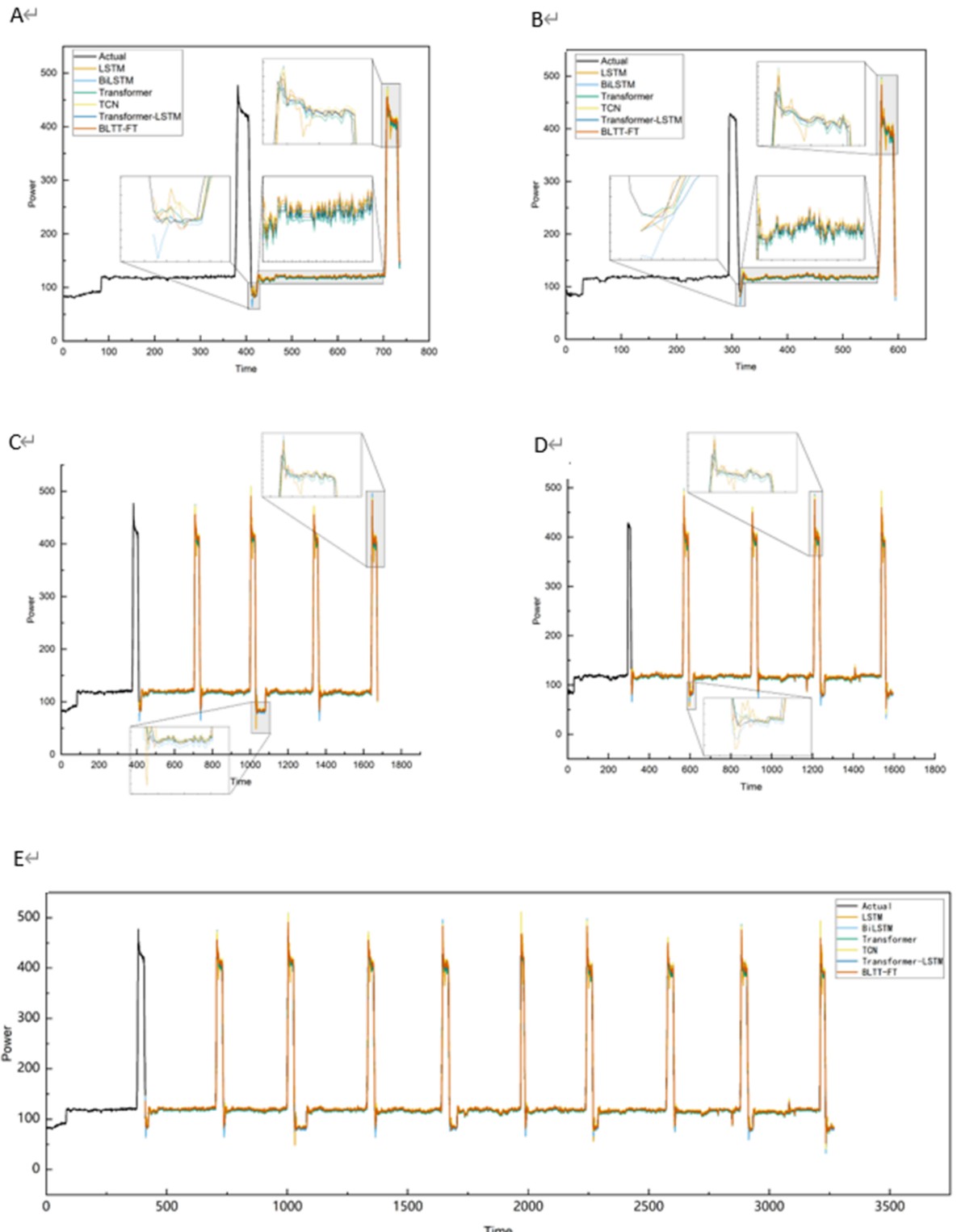

**Fig 15. Comparison of prediction results for different model comparisons with different prediction ranges for dataset 1.** A: Longitudinal direction at step 1. B: Latitudinal direction at step 1. C: Longitudinal direction at step 4. D: Latitudinal direction at step 4. E: Overall at step 9.

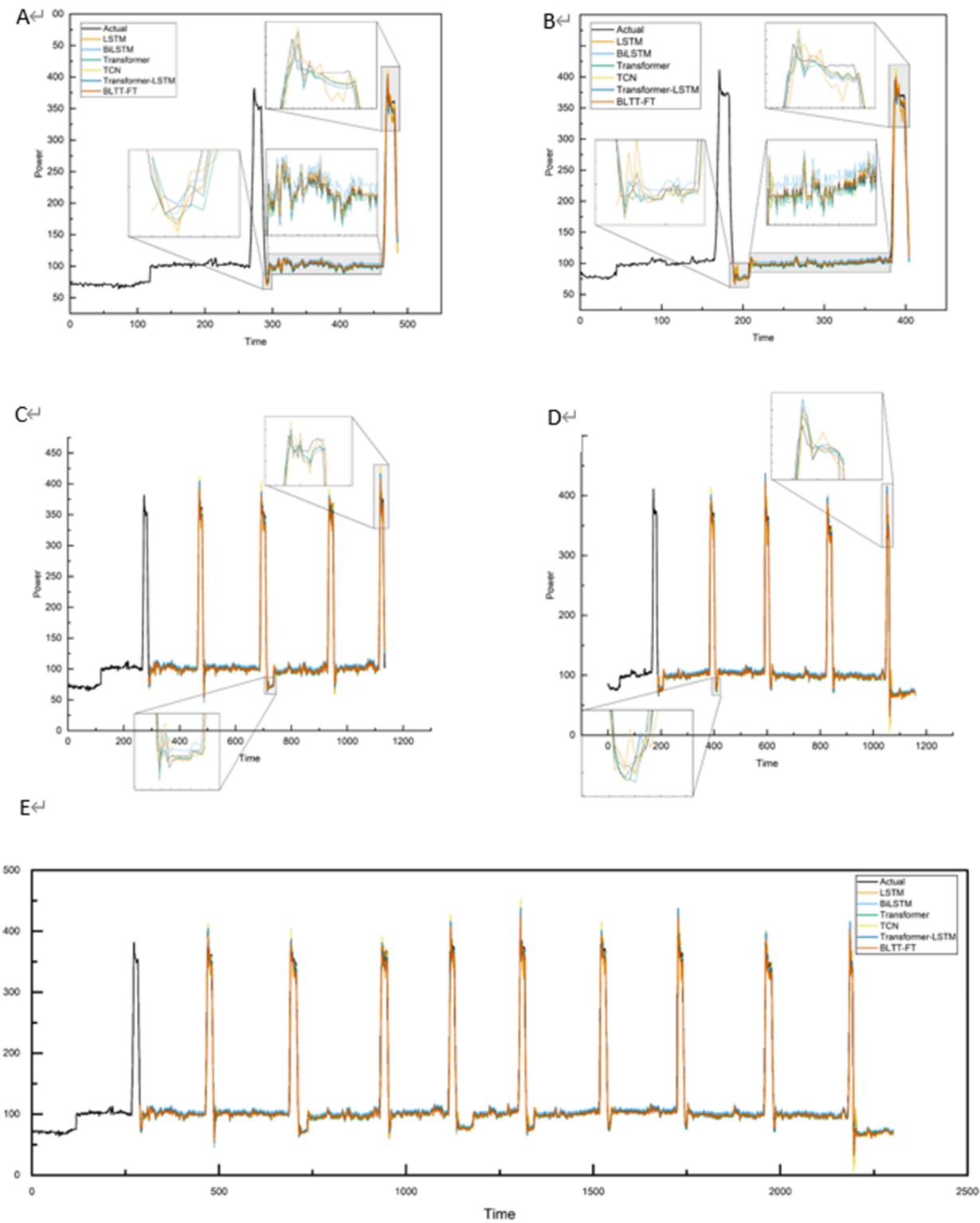

**Fig 16. Comparison of prediction results for different model comparisons with different prediction ranges for dataset 2.** A: Longitudinal direction at step 1. B: Latitudinal direction at step 1. C: Longitudinal direction at step 4. D: Latitudinal direction at step 4. E:Overall at step 9.

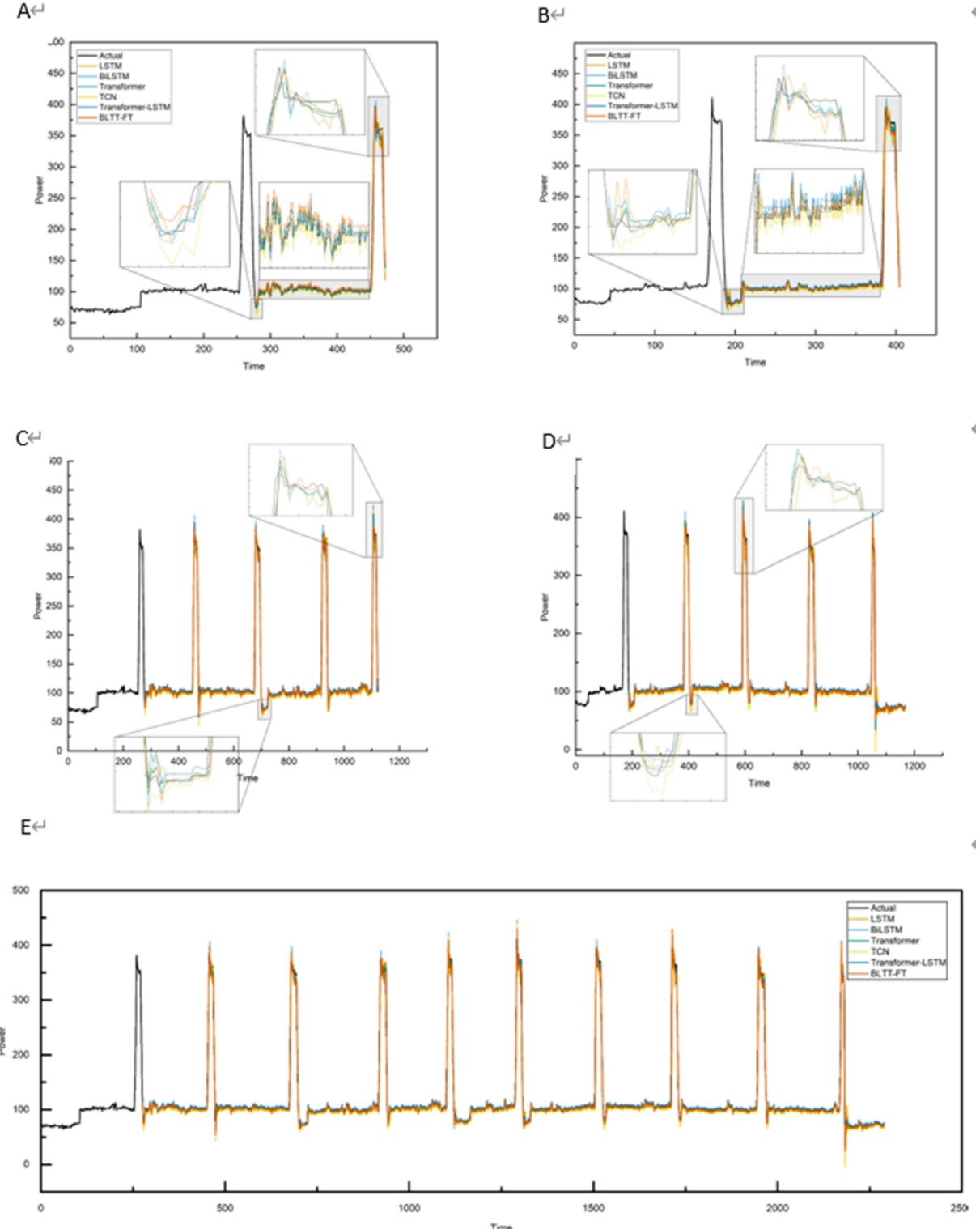

**Fig 17. Comparison of prediction results for different model comparisons with different prediction ranges for dataset 3.** A: Longitudinal direction at step 1. B: Latitudinal direction at step 1. C:Longitudinal direction at step 4. D: Latitudinal direction at step 4. E: Overall at step 9.

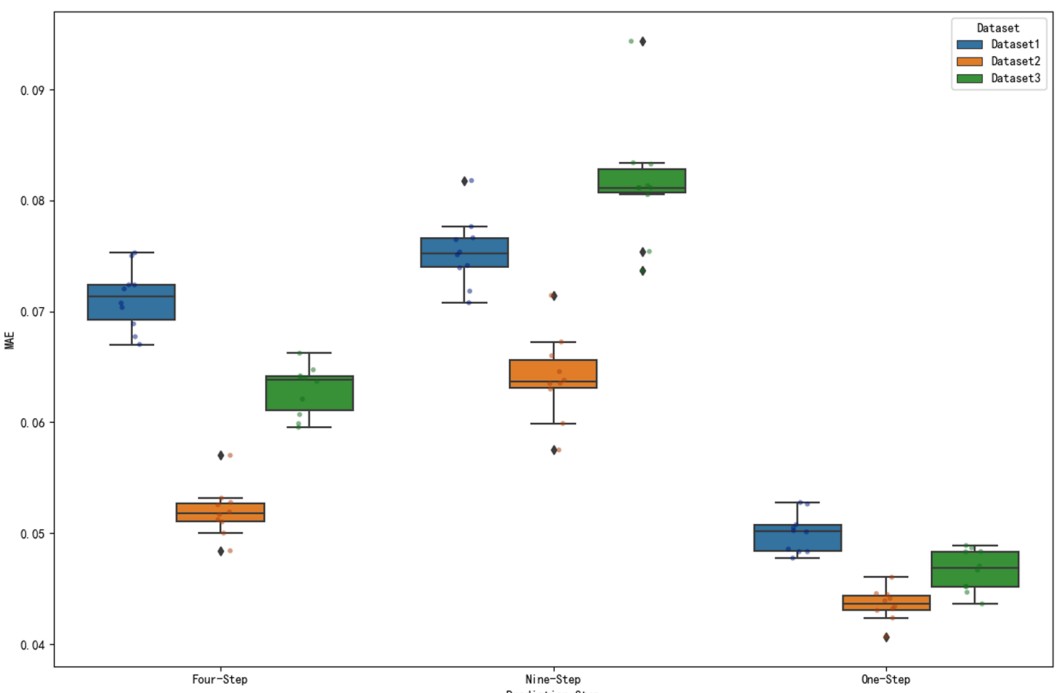

**Fig 18**. **Comparison of MAE at prediction step for different datasets.** The figure shows the distribution of MAE at steps 1, 4 and 9 for the three datasets.

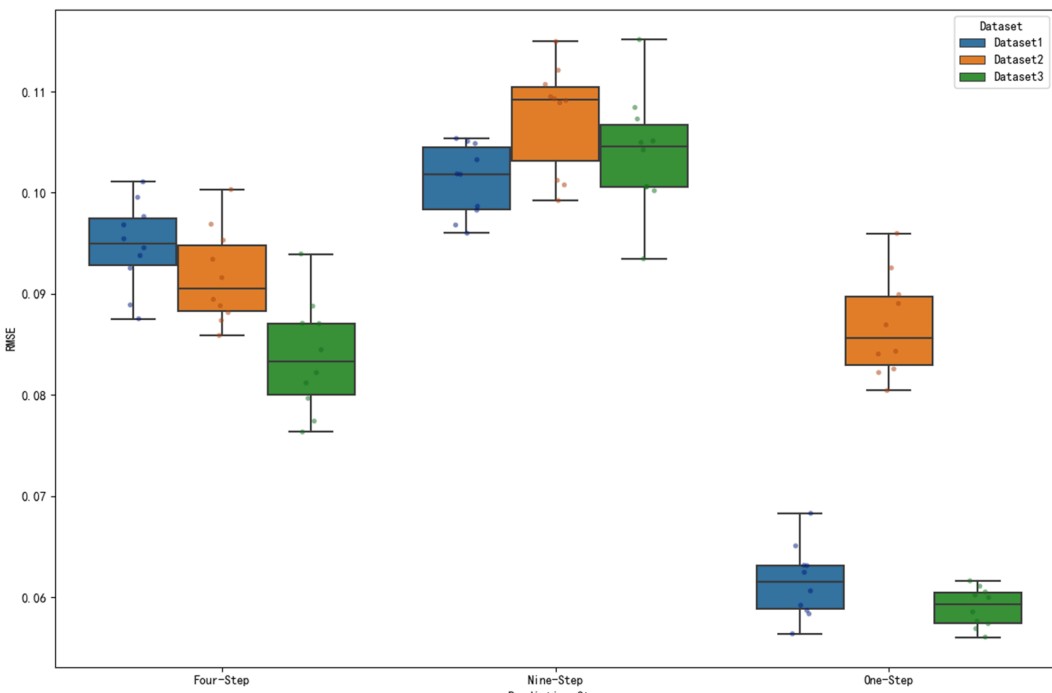

**Fig 19**. **Comparison of RMSE at prediction step for different datasets.** The figure shows the distribution of RMSE at steps 1, 4 and 9 for the three datasets.

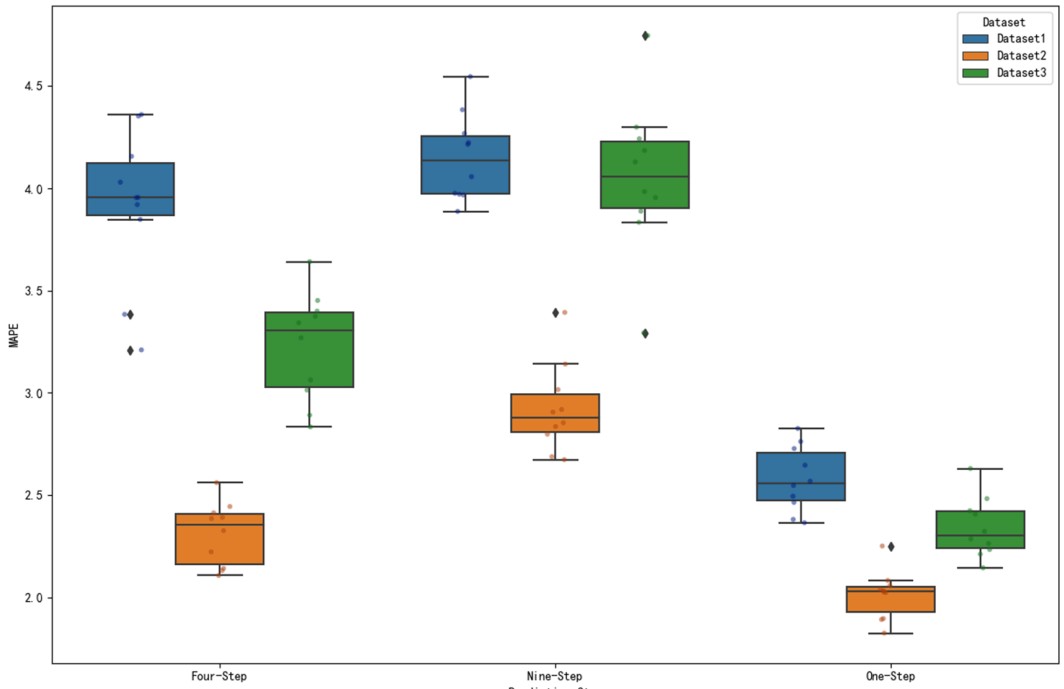

**Fig 20**. **Comparison of MAPE at prediction step for different datasets.** The figure shows the distribution of MAPE at steps 1, 4 and 9 for the three datasets.

**Table 9**. **Display of significance test results.**

| Norm | Predicted Step Size | F-value | P-value | significance Of Differences |
|------|--------------------|---------|---------|-----------------------------|
| MAE | 1 step | 35.3612 | 0.0001 | significant |
| MAE | 4 steps | 155.0869 | 0.0000 | significant |
| MAE | 9 steps | 43.4282 | 0.0000 | significant |
| RMSE | 1 step | 170.5940 | 0.0000 | significant |
| RMSE | 4 steps | 13.5924 | 0.0001 | significant |
| RMSE | 9 steps | 4.1268 | 0.0273 | significant |
| MAPE | 1 step | 39.8406 | 0.0000 | significant |
| MAPE | 4 steps | 83.6908 | 0.0000 | significant |
| MAPE | 9 steps | 60.0360 | 0.0001 | significant |

This table presents the significance test F-value, p-value and difference in significance for different indicators at different prediction steps.

Based on the results of the ablation experiments on the three datasets, the BLTT-FT model demonstrates significant performance advantages over single-module and two-module combination models in different prediction steps (1, 4, and 9 steps), which verifies the key role of multi-module synergy in capturing multi-step timing features. In the 1-step prediction of Dataset1, the MAE of the full model (0.0489 mm) is reduced by 57.9%, 53.4%, and 50.3% compared with Trans-TCN (0.1162 mm), Bi-LSTM-TCN (0.1051 mm), and Bi-LSTM-Transformer (0.0984 mm), respectively, showing that the immediate effectiveness of bidirectional temporal modeling with local feature extraction; while in 9-step prediction, the MAE of the full model (0.0762 mm) is 57.0% lower than that of Trans-TCN (0.1775 mm), Bi-LSTM-TCN (0.1452 mm), and Bi-LSTM-Transformer (0.1112 mm), respectively, 47.5%, and 31.5%, respectively, highlighting the irreplaceable nature of the global attention mechanism for long-distance dependent modeling.

**Table 10**. **Model naming.**

| Model | Alias | Combination | Parameterization |
|---|---|---|---|
| BLTT-FT | Model 1 | full model | Batch size=32, Sequence length=[1,4,9], Epoch=100, Initial learning rate=0.00001, Loss function=MSE, Activation function=ReLU; Bi-LSTM: Hidden size=64, Num_layers=1, Dropout=0.5; Transformer encoder: Encoder layers=6, Num_heads=8, Dropout=0.5; TCN: Layer neurons=32, Kernel size=7, Hidden layers=4, Activation=ReLU |
| Trans-TCN | Model 2 | Transformer+TCN | Transformer Encoder: number of encoder layers=6, number of heads=8, shedding=0.5; TCN: number of layer neurons=32, kernel size=7, number of hidden layers=4, activation=ReLU, batch size=32, sequence length=[1,4,9], Epoch=100, learning rate=0.00001 |
| Bi-LSTM-TCN | Model 3 | Bi-LSTM+TCN | Bi-LSTM: hidden size=64, number of layers=1, shedding=0.5; TCN: number of layer neurons=32, kernel size=7, number of hidden layers=4, activation=ReLU; batch size=32, sequence length=[1,4,9], Epoch=100, learning rate=0.00001 |
| TCN | Model 4 | single module | Kernel_size=3, N_levels=2, N_hid=32, Dilation_factor=[1,2], Dropout=0.5 |
| Bi-LSTM | Model 5 | single module | Hidden_size=64, Num_layers=2, Dropout=0.5 |
| Bi-LSTM-Transformer | Model 6 | Bi-LSTM+Transformer | Bi-LSTM: Hidden size=64, Num_layers=1, Dropout=0.5; Transformer Encoder: Encoder layers=6, Num_heads=8, Dropout=0.5; Batch size=32, Sequence length=[1,4,9], Epoch=100, Learning rate=0.00001 |
| Transformer | Model 7 | single module | Num_layers=1, Num_heads=8, Dropout=0.5 |

This table lists the different models and their aliases and combinations.

**Table 11**. **Comparison of test results (Dataset 1).**

| Dataset | Model | Evaluation Metrics | Forecast Range | | |
|---|---|---|---|---|---|
| | | | 1 step | 4 steps | 9 steps |
| 1 | **BLTT-FT** | MAE/mm | **0.0489** | **0.0741** | **0.0762** |
| | | RMSE/mm | **0.0625** | **0.0960** | **0.1001** |
| | | MAPE/% | **2.58** | **3.93** | **4.12** |
| | | $R^2$ | **0.9823** | **0.9437** | **0.9211** |
| | Trans-TCN | MAE/mm | 0.1162 | 0.1621 | 0.1775 |
| | | RMSE/mm | 0.1521 | 0.2170 | 0.2423 |
| | | MAPE/% | 6.32 | 8.81 | 9.75 |
| | | $R^2$ | 0.8621 | 0.8432 | 0.7951 |
| | Bi-LSTM-TCN | MAE/mm | 0.1051 | 0.1254 | 0.1452 |
| | | RMSE/mm | 0.1356 | 0.1651 | 0.1858 |
| | | MAPE/% | 5.54 | 6.81 | 7.85 |
| | | $R^2$ | 0.9111 | 0.8854 | 0.8521 |
| | TCN | MAE/mm | 0.1134 | 0.1629 | 0.1782 |
| | | RMSE/mm | 0.1492 | 0.2164 | 0.2421 |
| | | MAPE/% | 6.05 | 8.84 | 9.74 |
| | | $R^2$ | 0.8890 | 0.8391 | 0.7893 |
| | Bi-LSTM | MAE/mm | 0.1151 | 0.1654 | 0.1851 |
| | | RMSE/mm | 0.1502 | 0.2201 | 0.2555 |
| | | MAPE/% | 6.22 | 9.20 | 10.21 |
| | | $R^2$ | 0.8825 | 0.8230 | 0.7722 |
| | Bi-LSTM-Transformer | MAE/mm | 0.0984 | 0.1063 | 0.1112 |
| | | RMSE/mm | 0.1261 | 0.1361 | 0.1435 |
| | | MAPE/% | 5.31 | 5.75 | 6.05 |
| | | $R^2$ | 0.9013 | 0.8949 | 0.8890 |
| | Transformer | MAE/mm | 0.1196 | 0.1632 | 0.1767 |
| | | RMSE/mm | 0.1540 | 0.2192 | 0.2414 |
| | | MAPE/% | 6.37 | 8.82 | 9.80 |
| | | $R^2$ | 0.8424 | 0.8396 | 0.7887 |

The table shows the evaluation metrics such as MAE for different models on dataset 1 under 1 step, 4 steps, and 9 steps prediction in the ablation experiments, which are used to validate the modular contribution and combined effect of the BLTT-FT model.

**Table 12**. Comparison of test results (Dataset 2).

| Dataset | Model | Evaluation Metrics | Forecast Range | | |
|---|---|---|---|---|---|
| | | | 1 step | 4 steps | 9 steps |
| 2 | **BLTT-FT** | MAE/mm | **0.0439** | **0.0521** | **0.0636** |
| | | RMSE/mm | **0.0865** | **0.0931** | **0.1079** |
| | | MAPE/% | **1.99** | **2.26** | **2.85** |
| | | $R^2$ | **0.9921** | **0.9854** | **0.9420** |
| | Trans-TCN | MAE/mm | 0.0854 | 0.1101 | 0.1304 |
| | | RMSE/mm | 0.1608 | 0.2305 | 0.2703 |
| | | MAPE/% | 4.02 | 6.01 | 7.07 |
| | | $R^2$ | 0.9303 | 0.8901 | 0.8604 |
| | Bi-LSTM-TCN | MAE/mm | 0.0722 | 0.0901 | 0.1157 |
| | | RMSE/mm | 0.1258 | 0.1557 | 0.1902 |
| | | MAPE/% | 3.32 | 4.21 | 5.33 |
| | | $R^2$ | 0.9505 | 0.9303 | 0.8951 |
| | TCN | MAE/mm | 0.1022 | 0.1475 | 0.1756 |
| | | RMSE/mm | 0.1845 | 0.2623 | 0.3117 |
| | | MAPE/% | 4.75 | 7.42 | 9.07 |
| | | $R^2$ | 0.9002 | 0.8188 | 0.8088 |
| | Bi-LSTM | MAE/mm | 0.0952 | 0.1655 | 0.1957 |
| | | RMSE/mm | 0.1852 | 0.2806 | 0.3302 |
| | | MAPE/% | 4.31 | 8.04 | 9.53 |
| | | $R^2$ | 0.9306 | 0.8102 | 0.8005 |
| | Bi-LSTM-Transformer | MAE/mm | 0.0607 | 0.0702 | 0.0853 |
| | | RMSE/mm | 0.1007 | 0.1153 | 0.1351 |
| | | MAPE/% | 2.75 | 3.17 | 3.83 |
| | | $R^2$ | 0.9753 | 0.9681 | 0.9350 |
| | Transformer | MAE/mm | 0.0898 | 0.1489 | 0.1749 |
| | | RMSE/mm | 0.1753 | 0.2671 | 0.3162 |
| | | MAPE/% | 4.13 | 7.47 | 9.10 |
| | | $R^2$ | 0.9211 | 0.8187 | 0.8087 |

The table shows the evaluation metrics such as MAE for different models on dataset 2 under 1 step, 4 steps, and 9 steps prediction in the ablation experiments, which are used to validate the modular contribution and combined effect of the BLTT-FT model.

In Dataset2, with the extension of the prediction step from 1 to 9 steps, the RMSE of the full model increases from 0.0865 mm to 0.1079 mm, which is only 24.7%, while the RMSE of Trans-TCN increases from 0.1608 mm to 0.2703 mm, which is 68.8%, indicating that the full model effectively suppresses the long series through the collaboration of multiple modules error accumulation in prediction. Similarly, in the 9-step prediction of Dataset3, the R² of the complete model (0.9124) is increased by 24.1%, 22.4% and 25.6% compared with that of Bi-LSTM (0.7352), Transformer (0.7457) and TCN (0.7267), which verifies the advantages of bi-directional temporal modeling and global-local feature fusion in the modeling of long and complex series. modeling advantages.

Cross-dataset comparisons show that BLTT-FT consistently maintains the lowest error and highest accuracy in multi-step prediction regardless of the prediction step length, e.g., in the 4-step prediction of Dataset2, its MAPE (2.26%) is reduced by 69.5% compared with that of the single-module TCN (7.42%), and in the 9-step prediction of Dataset3, its MAE (0.0791 mm) is only 35.9% of the single-module Transformer (0.2206 mm).

The experimental results show that the multi-module combination effectively solves the problems of detail accuracy in single-step prediction and long-range dependence in multi-step prediction through the hierarchical modeling of bidirectional time-series capturing, global attention correlation, and local feature extraction, and its synergistic effect is especially significant in multi-step prediction scenarios, which provides a more optimal solution for multi-scale prediction of time-series data.

**Table 13.** Comparison of test results (Dataset 3).

| Dataset | Model | Evaluation Metrics | Forecast Range | | |
|---|---|---|---|---|---|
| | | | 1 step | 4 steps | 9 steps |
| 3 | **BLTT-FT** | MAE/mm | **0.0472** | **0.0635** | **0.0791** |
| | | RMSE/mm | **0.0583** | **0.0842** | **0.1040** |
| | | MAPE/% | **2.34** | **3.17** | **4.26** |
| | | $R^2$ | **0.9851** | **0.9510** | **0.9124** |
| | Trans-TCN | MAE/mm | 0.1051 | 0.1502 | 0.1904 |
| | | RMSE/mm | 0.1705 | 0.2404 | 0.2901 |
| | | MAPE/% | 5.51 | 8.01 | 10.07 |
| | | $R^2$ | 0.9220 | 0.8721 | 0.8422 |
| | Bi-LSTM-TCN | MAE/mm | 0.0855 | 0.1101 | 0.1402 |
| | | RMSE/mm | 0.1401 | 0.1855 | 0.2200 |
| | | MAPE/% | 4.57 | 6.09 | 7.51 |
| | | $R^2$ | 0.9422 | 0.9125 | 0.8821 |
| | TCN | MAE/mm | 0.1395 | 0.2115 | 0.2546 |
| | | RMSE/mm | 0.1983 | 0.2941 | 0.3523 |
| | | MAPE/% | 8.87 | 12.99 | 15.60 |
| | | $R^2$ | 0.8350 | 0.7440 | 0.7267 |
| | Bi-LSTM | MAE/mm | 0.1302 | 0.1951 | 0.2355 |
| | | RMSE/mm | 0.1900 | 0.2951 | 0.3452 |
| | | MAPE/% | 5.85 | 9.04 | 12.58 |
| | | $R^2$ | 0.8954 | 0.8256 | 0.7352 |
| | Bi-LSTM-Transformer | MAE/mm | 0.0653 | 0.0801 | 0.0959 |
| | | RMSE/mm | 0.1052 | 0.1251 | 0.1454 |
| | | MAPE/% | 3.24 | 4.01 | 4.80 |
| | | $R^2$ | 0.9681 | 0.9484 | 0.9251 |
| | Transformer | MAE/mm | 0.1104 | 0.1736 | 0.2206 |
| | | RMSE/mm | 0.1695 | 0.2627 | 0.3268 |
| | | MAPE/% | 5.27 | 8.92 | 12.47 |
| | | $R^2$ | 0.9013 | 0.8352 | 0.7457 |

The table shows the evaluation metrics such as MAE for different models on dataset 3 under 1 step, 4 steps, and 9 steps prediction in the ablation experiments, which are used to validate the modular contribution and combined effect of the BLTT-FT model.

## Conclusion

This research is oriented to the field of textile manufacturing and intelligent quality control, targeting the technical bottlenecks of the traditional experimental monitoring means for fabric tearing performance testing, and making innovative breakthroughs at the level of theoretical methods and engineering applications. The research proposes a BLTT-FT hybrid neural network framework, which realizes the deep modeling of complex timing features through the mechanism of multi-component synergy: Bi-LSTM captures the bi-directional temporal dependencies of electric power sequences, models the full-sequence long-distance feature associations with the self-attention mechanism of the Transformer coding layer, and extracts the local hierarchical features of variable-length sequences and reduces the number of parameters of the model by combining with the temporal convolutional network with residual connections, which significantly improves the model generalization ability and training efficiency. The architecture breaks through the traditional single model in complex sequences. The architecture breaks through the limitations of the traditional single model in modeling complex temporal dependencies and cross-scale feature interactions, and constructs a collaborative prediction framework that integrates sequence modeling, global attention mechanism and causal convolution.

In the multi-step prediction performance validation, the model demonstrates significant technical advantages, especially in the 9-step prediction scenario.Compared with other models, the proposed model performs well in real-time trend prediction for datasets constructed with different elasticity level fabrics and still maintains a low error level in the long prediction

range, with an average RMSE of 0.0881 for multi-step prediction, an average MAE of 0.0609 for multi-step prediction, an average MAPE as low as 3.06% for multi-step prediction, and an average coefficient of determination ($R^2$) of multi-step prediction as high as 0.9572. Analysis of variance (ANOVA) shows that the same model has highly significant differences in MAE, RMSE, and MAPE metrics on different datasets and at each prediction step ($p < 0.001$), and the gradient degradation problem in deep network training is effectively avoided by the residual TCN structure. In addition, the study realizes for the first time the engineering integration of deep learning model and multi-source data-driven situational awareness system, and develops a real-time prediction module for fabric tearing experiments based on electric power parameters, which can complete the prospective prediction of up to 9 groups of subsequent experimental results with a single set of experimental data, reduce the repetition of experimental processes, and improve the detection efficiency. Ablation experiments show that the multi-module combination effectively solves the problems of detail accuracy of single-step prediction and long-range dependence of multi-step prediction through the hierarchical modeling of bidirectional time-series capture, global attention correlation, and local feature extraction, and the synergistic effect is especially significant in multi-step prediction scenarios, which provides a more optimal solution for multi-scale prediction of time-series data.

The result constructs a closed-loop solution from industrial data to intelligent decision-making, provides a technical paradigm with engineering applicability for intelligent quality control in textile manufacturing, and promotes the transformation of traditional experimental process to data-driven intelligent prediction mode.

## Shortcomings and prospects

Although this study has made significant progress in the field of real-time monitoring and prediction of fabric tearing experiments, this area is still full of opportunities and challenges, providing rich directions for future research. First, we plan to deepen the optimization of the model to further enhance the prediction accuracy and generalization ability of the model by integrating more types of sensor data, such as temperature, humidity and other environmental parameters. Because fabrics are easily affected by temperature and humidity during the experimental process and their physical properties will be changed, it is necessary to include environmental influences in future research to reduce the uncertainty impact of external factors on the experiment. In this paper, when the data preprocessing, only the noise signal of the sensor is smoothed, and more advanced signal filtering technology should be used in the subsequent research, we plan to introduce adaptive filtering algorithm, which can dynamically adjust the filter parameters according to the real-time characteristics of the noise signal, so as to remove the noise interference more efficiently, and at the same time, in the phase of data acquisition, increase the signal shielding measures to reduce the electromagnetic interference. Meanwhile, at the data acquisition stage, signal shielding measures are added to reduce the impact of electromagnetic interference. At the same time, we will also work on applying the BLTT-FT model to other key aspects of textile production, such as the optimization of experiments on stretching and pilling, as well as the prediction of faults and maintenance of textile machinery, with a view to upgrading the entire production chain in an intelligent way.

## Acknowledgments

We would like to give special thanks to Yu Feng for her help in the data collection process.

## Author contributions

**Conceptualization:** Yang Lu.

**Data curation:** Qingchun Jiao.

**Formal analysis:** Qingchun Jiao.

**Funding acquisition:** Yang Lu.

**Investigation:** Yifan Zhang, Yang Lu.

**Methodology:** Qingchun Jiao.

**Project administration:** Yang Lu, Bo He, Min Zhu.

**Resources:** Bo He, Min Zhu.

**Software:** Qingchun Jiao, Yifan Zhang.

**Supervision:** Bo He, Kuokuo Wang.

**Validation:** Min Zhu.

**Visualization:** Yifan Zhang.

**Writing – original draft:** Yifan Zhang.

**Writing – review & editing:** Yifan Zhang.

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
