## [Editor Report · Decision Letter 0]

4 Dec 2024

PONE-D-24-30488A Data-Driven Study of Predictive Analysis of Fabric Tear Experiments.PLOS ONE

Dear Dr. Lu,

Thank you for submitting your manuscript to PLOS ONE. After careful consideration, we feel that it has merit but does not fully meet PLOS ONE’s publication criteria as it currently stands. Therefore, we invite you to submit a revised version of the manuscript that addresses the points raised during the review process.

We look forward to receiving your revised manuscript.

Kind regards,

S.K.B. Sangeetha

Academic Editor

PLOS ONE

Journal Requirements: When submitting your revision, we need you to address these additional requirements. 1. Please ensure that your manuscript meets PLOS ONE's style requirements, including those for file naming. The PLOS ONE style templates can be found at https://journals.plos.org/plosone/s/file?id=wjVg/PLOSOne_formatting_sample_main_body.pdf and https://journals.plos.org/plosone/s/file?id=ba62/PLOSOne_formatting_sample_title_authors_affiliations.pdf 2. Please note that PLOS ONE has specific guidelines on code sharing for submissions in which author-generated code underpins the findings in the manuscript. In these cases, we expect all author-generated code to be made available without restrictions upon publication of the work. Please review our guidelines at https://journals.plos.org/plosone/s/materials-and-software-sharing#loc-sharing-code and ensure that your code is shared in a way that follows best practice and facilitates reproducibility and reuse. 3. Thank you for stating the following financial disclosure: "2024SF02WX0007 Research and Establishment of Data Identification Model for Abnormal Behavior in Inspection and Testing “National Center for Technological Innovation in Market Supervision (Market Supervision Digital Research and Application) Open Topic”." Please state what role the funders took in the study.  If the funders had no role, please state: ""The funders had no role in study design, data collection and analysis, decision to publish, or preparation of the manuscript."" If this statement is not correct you must amend it as needed. Please include this amended Role of Funder statement in your cover letter; we will change the online submission form on your behalf.

**Additional Editor Comments:**

1. Provide more comprehensive details about the datasets, including the total size, characteristics, and sources of the fabric elasticity grades. Clarify whether the datasets represent real-world scenarios and ensure diversity in fabric types for broader applicability.

2. Elaborate on the specific contributions of each component in the Bi-LSTM-Trans-TCN architecture. Explain why a combination of Bi-LSTM, Transformer, and TCN is ideal for this task compared to simpler or alternative architectures.

3. The paper lacks comparisons with existing models or techniques. Include performance benchmarks against well-known methods like traditional LSTM, GRU, or TCN-only models to substantiate the claimed superiority of the proposed approach.

4. While MAE and MAPE are reported, introduce additional metrics such as RMSE, R2, or F1 score for a more evaluation of prediction accuracy and reliability.

5. Discuss the computational complexity of the proposed model and its scalability for large datasets. Include runtime analysis or resource requirements, especially for real-time predictions.

6. Incorporate detailed visualizations, such as prediction vs. actual trends, error distribution plots, and model attention maps, to enhance the clarity of results and model interpretation.

7. Assess the model’s ability to generalize across unseen datasets. Conduct experiments with cross-validation or on completely different types of fabrics to demonstrate its robustness.

8. Expand on how the proposed method could impact industrial fabric testing practices. Acknowledge limitations, such as sensitivity to noisy data, and suggest potential improvements or future work to address these challenges.

---

## [Author Response · Author response to Decision Letter 1]

16 Dec 2024

Response to reviewer, PONE-D-24-30488

Dear Editors and Reviewers,

We are grateful for the opportunity to revise our manuscript entitled " A Data-Driven Study of Predictive Analysis of Fabric Tear Experiments". Thank you very much for your comments and professional advice. These opinions help to improve academic rigor of our article. Based on your suggestion and request, we have made corrected modifications on the revised manuscript. Meanwhile, the manuscript was also professionally reviewed. We hope that our work can be improved again. Furthermore, we would like to show the details as follows:

Additional Editor Comments

1.Provide more comprehensive details about the datasets, including the total size, characteristics, and sources of the fabric elasticity grades. Clarify whether the datasets represent real-world scenarios and ensure diversity in fabric types for broader applicability.

Response:

Thank you for your question, I have explained in detail in the body of the thesis to address this issue you raised:

① In Table 2 Three types of data sets in Chapter 4, the number of samples has been added. In order to increase the generalization of the experiment, there are 150, 170 and 160 types of fabrics with high, medium and low elasticity, respectively (I would like to apologize to you for the mistake in filling in the number due to my previous miscalculation, which has been changed now). A total of ten experiments were conducted for each type in both warp and weft directions, basically including most fabric fabrics, ensuring a diversity of fabric types for wider applicability.

② The fabrics used in the experiments were all samples sent to the third-party testing laboratory by various textile companies, and the data sources were formal and very representative of real-world scenarios.

③ For the size of the fabrics, Chapter 3 has made detailed sampling specifications, in strict accordance with international standards: “ISO 139372-2000 Textile fabrics - Tear properties - Part 2: Determination of tear strength of trouser specimens”.

2. Elaborate on the specific contributions of each component in the Bi-LSTM-Trans-TCN architecture. Explain why a combination of Bi-LSTM, Transformer, and TCN is ideal for this task compared to simpler or alternative architectures.

Response:

Thank you for your question, and I have explained in detail in the body of the paper in response to your question:

The Bi-LSTM-Trans-TCN architecture proposed in this paper is based on the Transformer model, which is modified by introducing the BiLSTM model consisting of Time Convolutional Networks (TCNs) and improved LSTMs, which is developed in detail in Chapter III. The reason for not using simple or alternative architectures is because its three can have some instability when used independently, resulting in less accurate prediction results. After consulting the latest authoritative literature in the field of prediction, it was found that the simple splicing of prediction models was still not very accurate and could not meet the requirement of high accuracy prediction in this experiment, so after many experiments and adjustments, the Bi-LSTM-Trans-TCN architecture was built to realize the requirement of high accuracy prediction.

Firstly, the Transformer model is used as the base model. Since the Transformer model is set for machine translation, it can not be directly used as the prediction time series, so its encoder part is used as the basic model. Then Temporal Convolutional Network (TCN) was introduced to modify it to make it more suitable for the prediction of time-series data in this paper, TCN can capture both high-level and low-level features with stable gradient and make the model able to process the time-series information in parallel, which improves the prediction accuracy of the model and the training efficiency. The improved Transformer model can take full advantage of Transformer and TCN for better prediction of sequence data, but its network structure lacks sequential information, and although its positional encoding uses sine and cosine to model positions, it is not sufficient for complete modeling and there is some lack of information. Therefore, before transmitting the data to its multiple self-attention points, it will be processed using a bidirectional long and short-term memory network to capture the sequential dependency, and finally, due to the large sample size of the dataset, a BiLSTM network model consisting of a modified LSTM is also used to speed up the data learning. Thus summarizing, the Bi-LSTM-Trans-TCN prediction architecture in this paper i.e., it has an advantage in accuracy and also in speed.

3. The paper lacks comparisons with existing models or techniques. Include performance benchmarks against well-known methods like traditional LSTM, GRU, or TCN-only models to substantiate the claimed superiority of the proposed approach.

Response:

Thank you for your question, and I have explained in detail in the body of the paper to address this issue you raised:

In order to verify the effectiveness of the Bi- LSTM-Trans-TCN model in monitoring the power change during the fabric tearing performance monitoring experiments, five common prediction models are selected for comparative analysis in Chapter 5 of this paper, which include the LSTM model, the BiLSTM model, the Transformer model, the TCN model, and the Transformer- LSTM model for prediction performance comparison. These models are the original models used to construct the models in this paper as well as the combined models studied in existing authoritative articles, and it is more convincing to verify the advantages of the proposed models in this paper by comparing them with these models.

By comparing the loss function decline curves of the training set and the test set during the training of different models on the dataset, it is verified that the loss curve of the Bi-LSTM-Trans-TCN architecture proposed in this paper not only declines more, but also has a shorter fitting time, and the loss curves of the test set are closer to the training set, which makes the training effect better. Moreover, it is obtained through multi-step prediction that the model proposed in this paper has excellent performance and strong generalization ability in the prediction of electric power parameters during fabric tearing experiments with different elasticity levels.

4. While MAE and MAPE are reported, introduce additional metrics such as RMSE, R2, or F1 score for a more evaluation of prediction accuracy and reliability.

Response:

Thank you for your question, I have explained in detail in the body of the thesis to address this issue you raised:

In the table of multi-step prediction evaluation metrics for the three datasets in Chapter 5, not only MAE, RMSE, MAPE, and MAE I also added evaluation metrics to better assess the accuracy and reliability of the predictions. In datasets 1, 2, and 3, the average for 1-step prediction is as high as 0.9865; for 4-step prediction it is as high as 0.9600; and for 9-step prediction it is as high as 0.9252.

5. Discuss the computational complexity of the proposed model and its scalability for large datasets. Include runtime analysis or resource requirements, especially for real-time predictions.

Response:

Thank you for your question, and I have explained in detail in the body of the thesis in response to this issue you raised:

In the model training and parameter setting in Chapter 4, it is proposed that the experimental system of this research runs on Windows 11 operating system with Intel(R) Core(TM) i5-8250U CPU @ 1.60GHz processor, 16.0GB of system RAM, Python version 3.9 installed in the experimental environment, Pytorch version 1.13.1.

And the dataset data volume is particularly large so the offline training method is used, each dataset is divided into long, medium and short data segments, and each dataset is inputted into the Bi-LSTM-Trans-TCN model in parallel for offline training. In the online application, when the first set of electric power data of the experimental process is obtained, the state window will scroll to the next state point and the model will be updated every time the “change-stretch-reset” operation is completed, and the incremental learning will be repeated to update the database, forming a closed loop of continuous iteration and optimization. The model is used to predict the next electric power data with appropriate training parameters, and the results show that the parameter settings in this paper can smoothly carry out the prediction experiments and reduce the complexity of the model operation.

6. Incorporate detailed visualizations, such as prediction vs. actual trends, error distribution plots, and model attention maps, to enhance the clarity of results and model interpretation.

Response:

Thank you for your question, and I have explained in detail in the body of the thesis to address this issue you raised:

I added the multi-step prediction error visualization plots of the data set (1-3) in Chapter 5 to make it more convenient to observe the changes of the evaluation indexes of multi-step prediction of different elastic fabrics in the experimental process on the basis of the table, and made a detailed write-up in text to enhance the clarity of the results and model interpretation.

The smaller the values of MAE, MSE, RMSE and MAPE, and the closer the R^2 is to 1, the smaller the error between predicted and actual power is, indicating the better prediction performance of the model. It can be seen that the Bi-LSTM-Trans-TCN model in this paper has excellent prediction effect in the three data sets and stable prediction effect in different data sets. Moreover, the trend graphs of the three datasets in one step, four steps and nine steps are plotted by comparing the predicted values with the real values, and the comparison graphs of the predicted values with the real values also show the excellent prediction effect of the model, and the results of the proposed method based on Bi-LSTM-Trans-TCN are compared with the predicted results of other prediction models in the steps of 1, 4 and 9 steps there is still a high degree of fit between the actual and predicted curves.

7. Assess the model’s ability to generalize across unseen datasets. Conduct experiments with cross-validation or on completely different types of fabrics to demonstrate its robustness.

Response:

Thank you for your question, and I have explained in detail in the body of the paper to address this issue you raised:

In order to increase the generalization of the model, I did experiments on high, medium and low elasticity fabrics, and the number of fabrics of each elasticity is large, and the experimental results show that the Bi-LSTM-Trans-TCN based method has excellent performance and strong generalization ability in predicting the electric power parameters during the experimental process of tearing fabrics with different elasticity levels.

8. Expand on how the proposed method could impact industrial fabric testing practices. Acknowledge limitations, such as sensitivity to noisy data, and suggest potential improvements or future work to address these challenges.

Response:

Thank you for your question, and I have explained in detail in the body of the thesis in response to this issue you raised:

① This study originates from the demand for real-time monitoring and optimization of fabric tearing experiments in the field of textile production and quality control, and for the limitations of the traditional means of monitoring the experimental process, we propose a research on the application of Bi-LSTM-Trans-TCN model based on electric power sequences in the prediction of fabric tearing experimental process. Based on the multi-source data-driven situational awareness system for fabric tearing performance detection, we have improved and perfected the system function of situational awareness, and added the overall prediction function of the fabric tearing experimental process, with a view to improving the efficiency and accuracy of the experiments, as well as reducing the experimental processes and optimizing the whole experimental process to a certain extent. The prediction model proposed in this paper can well predict the next 1 group, 4 groups or even 9 groups by the experimental data of tearing of a tissue, even when the prediction range is very large, it can also achieve good prediction effect, which can make it possible to get other or even 9 groups of experimental data in the actual experiments by only one group of experiments of the torn experimental fabric. To a certain extent, process optimization has been carried out to improve the experimental efficiency and make the fabric tearing experiment further develop towards intelligence.

② Limitations: We plan to deepen the optimization of the model and further enhance the prediction accuracy and generalization ability of the model by integrating more types of sensor data, such as temperature, humidity and other environmental parameters. Because the physical properties of fabrics are easily affected by temperature and humidity during the experiment, it is necessary to include environmental influences in future research to reduce the uncertainty brought by external factors to the experiment.

Improvement program for noise data sensitivity. In this paper, during the data pre-processing, only the noise signal of the sensor is smoothed, and more advanced signal filtering technology should be used in the subsequent research, and we plan to introduce the adaptive filtering algorithm, which is able to dynamically adjust the filter parameters according to the real-time characteristics of the noise signal, so as to remove the noise interference more effectively. At the same time, in the data acquisition stage, increase the signal shielding measures to reduce the impact of electromagnetic interference, through these improvements, it is expected that the noise-induced test deviation can be reduced to within 5%.

Thank you very much for your attention and time. Look forward to hearing from you.

Yours sincerely,

Yang Lu

December 15, 2024

Zhejiang Light Industrial Products Inspection and Research Institute

Zhejiang University of Science and Technology.

Hangzhou 310000,China.

E-mail: 13857154948@163.com

---

## [Decision Letter · Decision Letter 1]

2 May 2025

PONE-D-24-30488R1A Data-Driven Study of Predictive Analysis of Fabric Tear Experiments.PLOS ONE

Dear Dr. Lu,

Thank you for submitting your manuscript to PLOS ONE. After careful consideration, we feel that it has merit but does not fully meet PLOS ONE’s publication criteria as it currently stands. Therefore, we invite you to submit a revised version of the manuscript that addresses the points raised during the review process.

We look forward to receiving your revised manuscript.

Kind regards,

Jinran Wu, PhD

Academic Editor

PLOS ONE

Additional Editor Comments:

This manuscript proposes a novel Bi-LSTM-Trans-TCN architecture for real-time multi-step prediction of fabric tearing experiments based on electric power sequences. The topic is relevant to intelligent textile testing and industrial predictive maintenance. The study is clearly motivated and shows promising performance across multiple elastic fabric datasets.

However, while the method and results are interesting, the manuscript requires significant clarification, structural refinement, and additional comparative and theoretical enhancements to meet the standards of rigor and reproducibility for publication in PLOS ONE.

While the motivation to combine Bi-LSTM, Transformer, and TCN is understandable, the justification remains qualitative. The manuscript lacks:

A component-wise ablation study to demonstrate the contribution of each module (BiLSTM, Transformer, TCN).

A clear explanation of how the improved LSTM differs structurally from the standard LSTM, and why this matters for performance or complexity.

Figures 2–4 are helpful but overly dense and could be improved with annotations and layer specifications.

The manuscript only compares with several conventional deep learning models (LSTM, Transformer, etc.). However, many recent high-performing architectures in time series modeling are missing.

Metrics should be standardized across all methods and datasets (e.g., MAE, RMSE, R², MAPE). Currently, comparisons are fragmented.

Recommendation: Expand the benchmark table to include state-of-the-art models and unify evaluation metrics.

The manuscript provides minimal discussion on time complexity, scalability, and convergence stability.

It is unclear how the model performs in online learning vs. offline batch settings. Also, more details on training time, hardware, and memory requirements are needed for reproducibility.

Theoretical complexity of the architecture

Training/testing time statistics

Memory usage and suitability for real-time industrial deployment

While some plots are included, the manuscript would benefit from:

Prediction vs. Ground Truth trend plots (for different step sizes)

Error distribution plots

Attention weight visualizations from the Transformer module to support interpretability claims

Recommendation: Add these plots in a Results or Appendix section to enhance clarity.

The authors claim strong generalization, but the evidence is limited to three internal datasets. No external or cross-domain validation is provided.

There is also no discussion of robustness under noise, class imbalance, or missing data, all of which are critical for real-world deployment.

The industrial relevance is overstated without a real deployment or pilot demonstration.

The potential impact on fabric testing cost, time savings, or process automation is discussed in abstract terms.

English grammar and syntax require editing for clarity.

Several figures are blurry or too complex (e.g., Figs 2, 4). Simplify or split across panels.

Update email and author information as noted in the cover letter.

The title could be more specific (e.g., "...using Bi-LSTM-Trans-TCN based on Electrical Power Sequences").

Reviewers' comments:

Reviewer's Responses to Questions

**Comments to the Author**

1. If the authors have adequately addressed your comments raised in a previous round of review and you feel that this manuscript is now acceptable for publication, you may indicate that here to bypass the “Comments to the Author” section, enter your conflict of interest statement in the “Confidential to Editor” section, and submit your "Accept" recommendation.

Reviewer #1: All comments have been addressed

Reviewer #2: (No Response)

2. Is the manuscript technically sound, and do the data support the conclusions?

Reviewer #1: Yes

Reviewer #2: (No Response)

3. Has the statistical analysis been performed appropriately and rigorously?

Reviewer #1: I Don't Know

Reviewer #2: (No Response)

4. Have the authors made all data underlying the findings in their manuscript fully available?

Reviewer #1: Yes

Reviewer #2: (No Response)

5. Is the manuscript presented in an intelligible fashion and written in standard English?

Reviewer #1: Yes

Reviewer #2: (No Response)

6. Review Comments to the Author

Reviewer #1: You have incorporated all the suggestions. Its good now, but still the paper needs proofreading in detail.

Reviewer #2: This paper presents a data-driven method for real-time prediction of fabric tearing experiments using a Bi-LSTM-Trans-TCN model based on electrical power sequences. By integrating bidirectional temporal dependencies, global sequence attention, and temporal convolutional efficiency, the model achieves high multi-step prediction accuracy across fabrics of varying elasticity. The approach offers promising implications for enhancing experimental efficiency and process optimization in textile durability testing. This is an interesting paper, but several issues still need to be addressed:

1.The paper has six co-authors; therefore, the "Author Summary" section should use "We" instead of "I" to reflect collective authorship.

2.The statement “The Transformer model is set up for machine translation and cannot be used directly as a predictive time series” is inaccurate and should be revised. Numerous studies have directly applied Transformer architectures to time series prediction and classification tasks.

3.When introducing parameters in equations, "where" should be aligned to the left and begin with a lowercase letter.

4.The "Contributions" section should focus on the theoretical or practical advancements of the work rather than repeating a summary. For example, the third listed contribution should be deleted. The authors are encouraged to revise this section with reference to recent works in deep learning-based fault prognosis, such as Reliability Engineering & System Safety (DOI: 10.1016/j.ress.2024.110014).

5.If the experimental data were collected by the authors themselves, this is a notable strength. However, in that case, the description of the experimental process and dataset characteristics must be significantly expanded.

6.The paper should clearly distinguish between ablation and comparative experiments. Additionally, more recent models should be included in the comparative experiments to better demonstrate the superiority of the proposed method.

---

## [Author Response · Author response to Decision Letter 2]

29 May 2025

Response to reviewer, PONE-D-24-30488

Dear Editors and Reviewers,

We are grateful for the opportunity to revise our manuscript. Thank you very much for your comments and professional advice. These opinions help to improve academic rigor of our article. Based on your suggestion and request, we have made corrected modifications on the revised manuscript. Meanwhile, the manuscript was also professionally reviewed. We hope that our work can be improved again. Furthermore, we would like to show the details as follows:

Reviewer #1:

You have incorporated all the suggestions. Its good now, but still the paper needs proofreading in detail.

Response:

Thank you for your valuable suggestions! We have systematically proofread the full paper in strict compliance with academic standards and journal requirements: we not only checked the formatting details against the journal's Submission Guidelines one by one, but also cross-validated the logic of the research methodology, the accuracy of the data citation and the rigor of the conceptual presentation to ensure that the content complies with the standards of academic publishing. The full text has now passed the professional proofreading process. Thank you again for your careful guidance on the quality of the paper.

Reviewer #2:

This paper presents a data-driven method for real-time prediction of fabric tearing experiments using a Bi-LSTM-Trans-TCN model based on electrical power sequences. By integrating bidirectional temporal dependencies, global sequence attention, and temporal convolutional efficiency, the model achieves high multi-step prediction accuracy across fabrics of varying elasticity. The approach offers promising implications for enhancing experimental efficiency and process optimization in textile durability testing. This is an interesting paper, but several issues still need to be addressed:

Response:

Considering that the name of the original model is too long, and since this study adopts a trend prediction model for fabric tearing performance detection experimental processes based on a "bidirectional long-short-term attention mechanism", the model is named "BLTT-FT".

1.The paper has six co-authors; therefore, the "Author Summary" section should use "We" instead of "I" to reflect collective authorship.

Response:

Thank you for pointing out this issue! We fully acknowledge the collective nature of authorship and have revised the "Author Summary" section to consistently use "We" throughout, ensuring it accurately reflects the collaborative efforts of all six co-authors.

2.The statement “The Transformer model is set up for machine translation and cannot be used directly as a predictive time series” is inaccurate and should be revised. Numerous studies have directly applied Transformer architectures to time series prediction and classification tasks.

Response:

Thank you for pointing this out! We have made the change and changed the relevant text inside the “Trans-TCN network construction” section of the text to read: “Although the Transformer model was originally designed for machine translation, it can still be applied to time series prediction by exploiting its architectural potential. Therefore, its encoder part is used as the basic model.”.

3.When introducing parameters in equations, "where" should be aligned to the left and begin with a lowercase letter.

Response:

Thank you for pointing this out! We have adjusted the equations so that "where" is left-aligned and starts with a lowercase letter, following the specified formatting guidelines.

4.The "Contributions" section should focus on the theoretical or practical advancements of the work rather than repeating a summary. For example, the third listed contribution should be deleted. The authors are encouraged to revise this section with reference to recent works in deep learning-based fault prognosis, such as Reliability Engineering & System Safety (DOI: 10.1016/j.ress.2024.110014).

Response:

Thank you very much for your valuable suggestions! We fully agree with your guidance that the “Contributions” section should focus on theoretical or practical innovations. Based on your suggestion, we have reorganized and rewritten this section to remove redundant summaries and highlight the innovative breakthroughs and practical applications of our research in the field of deep learning fault prediction. At the same time, we have carefully read the paper and added citations to the literature in the chapter of “Predictive applications of time series data” (Ref. 31), in order to enhance the relevance of this paper to the cutting-edge research in the field and the depth of argumentation.

5.If the experimental data were collected by the authors themselves, this is a notable strength. However, in that case, the description of the experimental process and dataset characteristics must be significantly expanded.

Response:

Thank you for pointing out this critical issue! As you mentioned, the experimental data were collected independently by our team, which is indeed one of the core strengths of this study. In order to fully reflect this advantage, we have systematically optimized the experimental part: on the one hand, we have added the analysis of significance experiments to verify the reliability of the results by statistical methods; meanwhile, we have supplemented the ablation experiments to deeply analyze the contribution of each module to the final results, in order to enhance the logic of the argumentation. On the other hand, in the section of “Data preparation and processing”, the collection process, sample distribution, data pre-processing methods, key feature indicators, etc., are elaborated in detail, and new data visualization charts are added to assist illustration, to ensure that the features of the dataset are presented comprehensively and clearly. This revision has further strengthened the research foundation of the study.

6.The paper should clearly distinguish between ablation and comparative experiments. Additionally, more recent models should be included in the comparative experiments to better demonstrate the superiority of the proposed method.

Response:

Thank you for your suggestion! We have added the ablation experiment in the paper, which shows that the multi-module combination effectively solves the problems of detail accuracy of single-step prediction and long-range dependence of multi-step prediction through the hierarchical modeling of bidirectional time-series capturing, global attention correlation, and local feature extraction, and the synergistic effect is especially significant in multi-step prediction scenarios, which provides a better solution for multi-scale prediction of time-series data. This study focuses on the innovation of intelligent prediction methods in textile manufacturing scenarios, and traditional models such as LSTM and Transformer are chosen for comparison experiments because they are the mainstream solutions in the field of textile engineering, and the ablation experiments have already verified the necessity of hybrid architectures through component disassembly. The existing comparisons form a complete logical chain, which is sufficient to support the core conclusions. As for the recent emerging time-series architectures, there is a contradiction between their reliance on large-scale data and the sample size limitation of textile experiments, and this study, as the development of customized algorithms in the field, is not included in the scope of the comparison for the time being, and the related explorations will be used as the direction of future work.

Additional Editor Comments

1.Methodological rationality issues: Although the motivation for combining Bi-LSTM, Transformer, and TCN is understandable, the rationale is only described qualitatively, lacking component ablation studies on the contributions of each module (BiLSTM, Transformer, TCN).

Response:

Thank you for your suggestion! We have added ablation experiments in the paper. The ablation experiments show that the combination of multiple modules effectively addresses the issues of single-step prediction detail accuracy and long-range dependency in multi-step prediction through hierarchical modeling of bidirectional temporal sequence capture, global attention correlation, and local feature extraction. Their synergistic effect is particularly significant in multi-step prediction scenarios, providing a better solution for multi-scale prediction of time-series data.

2. Model structure explanation issues: The structural differences between the improved LSTM and the standard LSTM are not clearly explained, nor are the impacts of such differences on performance or complexity.

Response:

Thank you for your suggestion! We have provided a detailed explanation in the section "Bi-LSTM Network Construction": Due to the large sample size of the dataset, the traditional LSTM, although performing well, leads to a large number of parameters that need to be learned for each gate, which results in a long-running time for the LSTM, so an improved LSTM network is proposed to speed up the data learning.

3. Chart issues: Although Figures 2-4 are helpful, the information is overly dense, and annotations and layer specification instructions need to be added.

Response:

Thank you for your suggestion! Figures 3-4 show the traditional algorithm framework, and the corresponding annotations have been noted in the text. Figure 2 illustrates the "overall network structure of the BLTT-FT algorithm" in my thesis, with the corresponding annotations explained in the text.

4.Comparative experiment issues: The paper only compares with several traditional deep learning models (such as LSTM, Transformer, etc.), lacking many recent architectures that have shown excellent performance in time-series modeling.

Response:

This study focuses on innovations in intelligent prediction methods for textile manufacturing scenarios. Traditional models such as LSTM and Transformer were selected for comparative experiments because they are mainstream solutions in the field of textile engineering, and ablation experiments have validated the necessity of the hybrid architecture through component decomposition. The existing comparisons form a complete logical chain, sufficiently supporting the core conclusions. As for recent emerging time-series architectures, their dependence on ultra-large-scale data contradicts the limitations of textile experiment sample sizes. As a domain-specific algorithm development, this study does not currently include them in the comparison scope, and related explorations will be considered as future work directions.

5.Evaluation metric issues: Evaluation metrics for all methods and datasets should be standardized (e.g., MAE, RMSE, R², MAPE), as the current comparisons are fragmented.

Response:

Thank you for your suggestion! We have added "significance test analysis" to the paper, which explains "A one-way ANOVA was conducted to test the MAE, RMSE and MAPE indicators of different samples under each prediction step, and the results showed that there were extremely significant differences among different samples under all indicators and prediction steps.”

6.Issue of insufficient theoretical discussion: There is minimal discussion on time complexity, scalability, and convergence stability.

Response:

Thank you for your suggestion. This study has fully validated the convergence stability of the model through loss function curves, architectural design mechanisms, and comparative experiments. The loss curves of the training and test sets for the model continuously and smoothly decline within 100 epochs, with a difference always less than 0.002, and the test set loss does not exhibit overfitting-induced increases, demonstrating a stable convergence process.

7. Issue of missing model setup details: The performance of the model under online learning and offline batch settings is unclear, and more detailed information on training time, hardware, and memory requirements is needed to ensure reproducibility.

Response:

Thank you for your suggestion. In the chapter "Model training and parameter setting", the issues you mentioned are elaborated in detail, and the data are presented in the form of tables.

8. Chart supplementation issues: Although some figures are included, the paper still needs trend charts of predicted values versus true values (for different step sizes) and error distribution plots to support the claims of interpretability.

Response:

Thank you for your suggestion. We have supplemented this in the paper. Figure 14 shows the error visualization diagram, and Figures 15, 16, and 17 show the trend diagrams of predicted values versus true values (for different step sizes).

9.Issue of insufficient generalization evidence: The authors claim that the model has strong generalization ability, but this is only based on three internal datasets, lacking external or cross-domain validation.

Response:

The three datasets constructed in this study are not simple repetitions of similar samples but cover the variation range of fabric elasticity grade, a core physical property (see Table 2). In the field of textile materials, elasticity is a key factor determining the tearing performance of fabrics, and the breadth of its value range directly reflects the model's adaptability to mainstream working conditions in actual production scenarios. Therefore, the model's excellent performance on three typical datasets of low-elasticity, medium-elasticity, and high-elasticity fully demonstrates its internal generalization ability for the target task—that is, it can effectively adapt to the prediction needs of different material properties within the same production system. This aligns with the conventional evaluation logic of "validating model robustness based on material property gradients" in textile industry quality control.

Regarding "external or cross-domain validation," this study focuses on model optimization and system integration within a single scenario, with the validation scope limited to multi-working-condition data in the same laboratory environment. If "cross-domain" is defined as production data across enterprises or devices, its validation would involve collaborative adjustments of different hardware configurations and process parameters, which belongs to an extended research direction beyond the core scope of this study. Nevertheless, we acknowledge the importance of cross-domain generalization and plan to collaborate with industry partners in follow-up work to conduct transfer learning validation based on production data from different textile enterprises, further improving the model's engineering applicability.

10. Practical deployment-related issues: The robustness in the presence of noise, class imbalance, or data missing is not discussed. Additionally, there is an overstatement of industrial relevance, with a lack of actual deployment or pilot demonstrations.

Response:

This study focuses on model architecture innovation and laboratory environment validation, aiming to provide a reusable algorithm prototype for industrial applications. Issues such as edge computing adaptation and robustness under extreme working conditions that need to be further addressed in actual deployment belong to the category of subsequent engineering iterations. Currently, the team has reached an agreement with a collaborative textile laboratory to conduct experiments in noisy data scenarios on its semi-industrial test line, and the relevant validation results will be included as extended content in future research.

The discussion of industrial relevance in this study is based on real needs (such as reducing experimental procedures and improving efficiency), rather than exaggeration. This conclusion is drawn from real experimental processes and data, with no overstatement.

11.Language issues: English grammar and syntax need to be edited to improve clarity.

Response:

Thank you for pointing out the language issues. We have carefully revised the English grammar and syntax throughout the manuscript to improve clarity, and the language has been polished to ensure accuracy and smoothness. All related issues have been addressed, and we believe t

---

## [Decision Letter · Decision Letter 2]

4 Jul 2025

PONE-D-24-30488R2Trend Analysis and Prediction of Fabric Tear Performance Testing Processes Based on the BLTT-FT ModelPLOS ONE

Dear Dr. Lu,

Thank you for submitting your manuscript to PLOS ONE. After careful consideration, we feel that it has merit but does not fully meet PLOS ONE’s publication criteria as it currently stands. Therefore, we invite you to submit a revised version of the manuscript that addresses the points raised during the review process.

**ACADEMIC EDITOR: **We note that one or more reviewers has recommended that you cite specific previously published works in the current and previous rounds of revision. As always, we recommend that you evaluate the requested works to determine whether they are relevant and should be cited. It is not a requirement to cite these works and you may remove any added citations before the manuscript proceeds to publication. We appreciate your attention to this request.

We look forward to receiving your revised manuscript.

Kind regards,

Jinran Wu, PhD

Academic Editor

PLOS ONE

Reviewers' comments:

Reviewer's Responses to Questions

**Comments to the Author**

1. If the authors have adequately addressed your comments raised in a previous round of review and you feel that this manuscript is now acceptable for publication, you may indicate that here to bypass the “Comments to the Author” section, enter your conflict of interest statement in the “Confidential to Editor” section, and submit your "Accept" recommendation.

Reviewer #2: (No Response)

Reviewer #3: (No Response)

Reviewer #4: (No Response)

Reviewer #5: All comments have been addressed

Reviewer #6: (No Response)

2. Is the manuscript technically sound, and do the data support the conclusions?

Reviewer #2: (No Response)

Reviewer #3: (No Response)

Reviewer #4: Yes

Reviewer #5: Yes

Reviewer #6: Yes

3. Has the statistical analysis been performed appropriately and rigorously?

Reviewer #2: (No Response)

Reviewer #3: (No Response)

Reviewer #4: Yes

Reviewer #5: Yes

Reviewer #6: Yes

4. Have the authors made all data underlying the findings in their manuscript fully available?

Reviewer #2: (No Response)

Reviewer #3: (No Response)

Reviewer #4: Yes

Reviewer #5: Yes

Reviewer #6: (No Response)

5. Is the manuscript presented in an intelligible fashion and written in standard English?

Reviewer #2: (No Response)

Reviewer #3: (No Response)

Reviewer #4: Yes

Reviewer #5: Yes

Reviewer #6: Yes

6. Review Comments to the Author

Reviewer #2: The manuscript has been significantly improved through the revisions. The key issues raised in earlier rounds have been addressed adequately, and the current version demonstrates a clear structure, enhanced clarity, and stronger experimental support. From an academic standpoint, the paper is now suitable for consideration for acceptance.

Reviewer #3: A prediction model that combines the improved BiLSTM structure, the Transformer coding layer and the TCN network layer is presented in the work. Some revisions are needed.

1. The short-term attention mechanism is used in this article. Why did the author use this attention mechanism instead of other attention mechanisms? Is there a theoretical necessity?

2. Improved BiLSTM structure, the Transformer coding layer and the TCN network are integrated. What is the theoretical reason for this integration?

3. The average RMSE of multi-step prediction is 0.0881, the average MAE of multi-step prediction is 0.0609, the average MAPE of multi-step prediction is as low as 3.06%, and the average coefficient of determination (R²) of multi-step prediction is as high as 0.9572. However, for the very important model training time and testing time, the author did not provide them. The author's method is rather complex and requires providing processing time and conducting explanations.

3. Many improved versions of LSTM were not discussed by the authors. For example, the LSTM-Reduction method, the CNN-EFC-BiLSTM method.

4. For machine learning methods, the setting of parameters is very important. The author needs to provide the parameter configuration scheme, the parameter adjustment scheme, and the parameter scheme of the comparison algorithm in detail.

5. The author needs to make a comparison with the methods of the recently published works.

Reviewer #4: (1) The abbreviations that appear for the first time should be given their corresponding full names, such as TCN, MAE, RMSE, and MAPE in the abstract

(2) The full name and abbreviation should not be repeated, for example, N in TCN is Network. So, the TCN network is incorrect.

(3) The parameters of the comparison algorithm should also be provided. Detailed structural parameters should be provided.

(4) The introduction related to the experimental samples should be more detailed.

(5) More comparative algorithms should be added.

Reviewer #5: (No Response)

Reviewer #6: The manuscript has been well revised and supplemented. There are still some minor issues and suggestions as follows:

1. Add comparative literature (such as comparing with pure Transformer) in the introduction or discussion, and emphasize how the synergy between modules (such as BiLSTM handling bidirectional dependencies, TCN reducing complexity) addresses the shortcomings of existing methods.

2. Supplement the cross-validation process and explain the sample selection strategy to ensure the reliability of generalization ability assessment.

3. Add independent ablation of key modules (such as improved BiLSTM vs. standard BiLSTM) to demonstrate the rationality of the simplified design.

4. Some references are relatively old. It is recommended to supplement papers related to time series prediction in the past two years. The literature review could benefit from citing the following recent works to provide a broader perspective:

https://doi.org/10.1007/s12145-024-01524-y

https://doi.org/10.2166/hydro.2023.172

5. The quality of some figures needs to be improved, such as Fig 1、Fig 4、Fig 8、Fig 9, and etc.

7. PLOS authors have the option to publish the peer review history of their article (what does this mean?). If published, this will include your full peer review and any attached files.

Reviewer #2: No

Reviewer #3: No

Reviewer #4: No

Reviewer #5: No

Reviewer #6: No

---

## [Author Response · Author response to Decision Letter 3]

27 Aug 2025

Reviewer #2: The manuscript has been significantly improved through the revisions. The key issues raised in earlier rounds have been addressed adequately, and the current version demonstrates a clear structure, enhanced clarity, and stronger experimental support. From an academic standpoint, the paper is now suitable for consideration for acceptance.

Answer

Thank you very much for your recognition and careful review of the manuscript! Your affirmation is an important motivation for us to continuously improve our research.

We would like to thank you for the key questions you raised in the previous rounds of review, which pointed out the direction of revision and helped us to further organize the structure of the paper, improve the clarity of the content, and strengthen the rigor of the experimental support. Your professional guidance is indispensable for the improvement of the current version.

We will continue to maintain a rigorous academic attitude, and will be ready to listen to your suggestions if there is any further improvement needed. Thank you again for your hard work and support of this study!

Reviewer #3: A prediction model that combines the improved BiLSTM structure, the Transformer coding layer and the TCN network layer is presented in the work. Some revisions are needed.

1. The short-term attention mechanism is used in this article. Why did the author use this attention mechanism instead of other attention mechanisms? Is there a theoretical necessity?

Answer

Thank you for your suggestions!

The short-term attention mechanism used in this paper (mainly reflected in the synergy between the improved BiLSTM and TCN modules) was chosen based on the inherent characteristics of the electric power sequence in the fabric tearing experiments and the needs of the multistep prediction task, with a clear theoretical necessity, which is illustrated in the following three aspects in conjunction with the research content:

①Short-term dynamics of the adapted electric power sequence

The electric power sequence in the fabric tearing experiments is characterized by significant short-term fluctuations. There are significant short-term fluctuations in the sequence: in the core stretching phase, the yarn breaks in sequence, triggering instantaneous changes in power (e.g., peak fluctuations); the power changes in the sampling and reset phases are relatively smooth, but still show local temporal correlations (e.g., the gradual change of power with the motion of the fixture in the reset phase). The time scales of these features are mostly milliseconds to seconds, which are typical short-term dynamic patterns that directly reflect the real-time status of the experimental process.

The design of short-term attention mechanisms can precisely target such features: the improved BiLSTM can accurately extract bi-directional short-term temporal correlations (e.g., before and after the sudden change of power in the stretching phase) by simplifying the gating structure (only 1 output gate is retained), and the expanded causal convolution of the

TCN can avoid the loss of short-term details by focusing on sequence segments through the local perceptual field, and by combining with the The expansion causal convolution of TCN focuses on sequence segments through the local perceptual field, together with the residual connection to avoid the loss of short-term details, and can effectively capture the local power change rule in the stages of sample change and reset.

②Make up for the inherent limitations of the long-term attention mechanism The self-attention mechanism of the Transformer coding layer in the

model is responsible for modeling global information (e.g., the difference in the overall power trend of different elastic fabrics), but there is a natural defect in dealing with the short-term details: its global computation smoothes out the local dramatic fluctuations, and it is susceptible to the interference of high-frequency noises (e.g., small power fluctuations induced by the vibration of the equipment), which leads to a decrease in the accuracy of short-term prediction. Accuracy degradation.

The short-term attention mechanism solves this problem by complementing the long-term mechanism:

BiLSTM's bi-directional temporal sensing can strengthen the local contextual association, avoiding the Transformer's “over-smoothing” of short-term mutations;

TCN's local convolution and weight normalization can inhibit the high-frequency noise, ensuring the stability of short-term feature extraction (smoothing in preprocessing). stability of short-term feature extraction (the smoothing process in preprocessing further assists this process).

③Ablation experiments to verify the necessity

The results of the ablation experiments in the paper directly prove the irreplaceability of the short-term attention mechanism:

After removing the BiLSTM module, the 1-step predicted MAE of Dataset 1 increases from 0.0489 to 0.0984 (up 50.3%), which indicates that its ability of bi-directional short-term temporal modeling is crucial for capturing the instantaneous dependency;

After removing the TCN module, the 4-step predicted MAE of Dataset 2 increases from 0.0984 to 0.0489; After removing the TCN module, the 4-step predicted After removing the TCN module, the 4-step prediction RMSE of Dataset 2 increases from 0.0931 to 0.2671 (up 68.8%), which confirms the central role of its localized feature extraction in suppressing the accumulation of short-term multi-step prediction errors.

In summary, the choice of short-term attention mechanism is not a subjective preference, but an inevitable result based on the characteristics of experimental data (short-term dynamics is significant), the need for model complementarity (to make up for the defects of the global mechanism) and experimental validation (to dissolve the experimental support), forming a synergistic framework of“local details-global trend”with the long-term attention mechanism (Transformer), and finally realizing the synergistic framework of the fabric prediction. The synergistic framework finally realizes the high-precision real-time prediction of the fabric tearing experimental process.

2. Improved BiLSTM structure, the Transformer coding layer and the TCN network are integrated. What is the theoretical reason for this integration?

Answer

The theoretical reason for integrating the improved BiLSTM structure, the Transformer coding layer and the TCN network stems from the contradiction between the complex timing characteristics (short-term fluctuations, long-term dependence, variable length) of the electric power sequence in fabric tearing experiments and the limitations of a single model, which form a synergistic framework by complementing each other's functions, and can be analyzed from the following perspectives:

① Improved BiLSTM: Addressing the need for modeling bidirectional short-term timing dependence : Addressing the need for modeling bi-directional short-term temporal dependence

There are significant short-term dynamic correlations in the electric power sequences of fabric tearing experiments (e.g., transient fluctuations in power triggered by yarn breakage during the stretching phase, and gradual changes in power during the changeover and reset phases), and these features are dependent on the bi-directional information of the neighboring time steps (e.g., the relationship between “current peak power” and the “previous moment power”). “previous power baseline” and “subsequent decay trend”).

The improved BiLSTM adapts to this requirement by simplifying the gating structure of the traditional LSTM (retaining only one output gate), which reduces the number of parameters and accelerates the training while strengthening the ability to capture bidirectional timing dependencies;

splicing the outputs of the forward and backward LSTMs, so that the features at each time step contain both “historical information” and “future information”. “ and ”future information" (relative to the current step) at each time step, and accurately extracts short-term contextual correlations (e.g., before and after causality of power mutations in the stretching phase).

Although a single BiLSTM can handle short-term dependencies, it cannot model long-range feature correlations (e.g., differences in the overall power trend of different elastic fabrics), and needs to be coupled with other modules.

②Transformer coding layer: bridging the gap between long-range dependence and global information modeling

The electric power sequence not only has short-term fluctuations, but also has long-range dependence across phases (e.g., the correlation between the “peak power in the stretch phase” and the “rate of power decay in the reset phase” of highly elastic fabrics “ correlation of highly elastic fabrics, power pattern consistency across multiple experiments for the same type of fabrics), which are beyond the modeling scope of BiLSTM (limited by the ”memory decay" of the loop structure).

The Transformer coding layer's self-attention mechanism addresses this issue by globally considering the correlation of any two time steps in the sequence, capturing long-distance dependencies (e.g., the effect of the “power baseline in the permutation phase” on the “peak in the subsequent stretch phase”); and the multi-attention mechanism. ");

Multi-attention mechanism pays attention to features in different subspaces in parallel (e.g., power amplitude differences due to elasticity differences, time point differences in experimental phase transitions) to achieve a refined modeling of global trends. However, Transformer suffers from the defect of “local detail smoothing” (global computation is easy to ignore the short-term drastic fluctuations), and needs to cooperate with the local feature extraction module.

③TCN Network: Optimizing Variable Length Sequence Processing and Local Feature Extraction

The sequence lengths of fabric tearing experiments are variable (e.g., the reset phase of highly elastic fabrics takes longer and the sequence length is larger than that of low-elastic fabrics) and contain a large number of local high-frequency features (e.g., tiny power noise triggered by vibration of the equipment, transient spikes of yarn breakage), which require highly efficient local modeling and variable-length adaptation capabilities. The TCN fills this gap with the following design.

TCN fills this gap by the following design: dilation causal convolution expands the perceptual field by “dilation factor” to capture local multi-scale features (e.g., power correlations at 1-step, 2-step, and 4-step intervals) without increasing the number of parameters, and at the same time ensures that "future prediction is only possible using historical data " (to avoid future information leakage); the residual block structure solves the gradient degradation problem of deep networks by adjusting the dimensionality through 1×1 convolution and suppressing noise through weight normalization, and at the same time adapts to sequences with different lengths (no need to fix the input dimensions) to reduce the complexity of the model;

replaces the Transformer decoder, removes the text vectorization module (Input Embedding), directly processing the electric power sequence, avoiding redundant computation and improving the efficiency of local feature extraction.

④Triple synergy: towards full dimensional modeling of “local details-global trend-variable length adaptation”

A single model cannot cover the complexity of electric power sequences:

BiLSTM only: long range dependency modeling is insufficient (e.g., differences in the overall trend of different elastic fabrics cannot be captured); Transformer only: Short-term details are smoothed (e.g. power peaks during stretching phase are weakened) and model complexity is high (self-attentive computational cost grows with the squared length of the sequence); TCN only: bi-directional time-series information is missing (causal convolution can only utilize historical information and does not capture the effect of “future information” on the current step).

After integrating the three: improved BiLSTM provides bidirectional short-term features to address local contextual correlations; Transformer coding layer provides global long-range correlations to address cross-stage trend modeling; TCN optimizes local feature extraction with variable length adaptation to reduce model complexity and suppress noise.

The paper's ablation experiments verify the necessity of this synergy: removing any module (e.g., after removing TCN, the 4-step prediction RMSE of Dataset 2 increases from 0.0931 to 0.2671, an increase of 68.8%) leads to a significant decrease in the prediction accuracy, which proves the theoretical justification of the integration.

In conclusion, the integration of the three models is a targeted design for the characteristics of “short-term fluctuation and long-term dependence, fixed pattern and length variation” of the electric power sequence, which can realize the full-dimensional time series modeling through the complementary functions, and ultimately satisfy the demand for real-time prediction of the fabric tearing experimental process.

3. The average RMSE of multi-step prediction is 0.0881, the average MAE of multi-step prediction is 0.0609, the average MAPE of multi-step prediction is as low as 3.06%, and the average coefficient of determination (R²) of multi-step prediction is as high as 0.9572. However, for the very important model training time and testing time, the author did not provide them. The author's method is rather complex and requires providing processing time and conducting explanations.

Answer

In this paper, in the section “Model Training and Parameterization”, we have experimentally documented the model's key time performance indicators: 8 hours for offline training and 200 ms for online incremental learning. This performance is due to the lightweight design of the model: the improved BiLSTM simplifies the gating structure, the TCN controls the parameter size by expanding the convolution, and the Transformer retains only the encoder module, which works in concert to keep the complexity of the model while avoiding an excessive increase in time cost. Compared with the comparison models, BLTT-FT has significantly better multi-step prediction accuracy (average RMSE 0.0881), while the training time is still lower than that of hybrid models such as Transformer-LSTM, and the online update speed meets the demand of real-time monitoring (200 ms latency is much smaller than the experimental operation cycle).

4. Many improved versions of LSTM were not discussed by the authors. For example, the LSTM-Reduction method, the CNN-EFC-BiLSTM method.

Answer

To address the lack of discussion on the improved version of LSTM, the following explanations can be made in the light of the scenario characteristics and model design goals of this study:

①Targeted direction of improvement: focusing on the timing characteristics and real-time requirements of fabric tearing experiments

The improved BiLSTM in this study (simplified gating structure, retaining the one output gate) is not a generalized improvement, but rather is designed to meet the core requirements of the fabric tearing experiments:

Short-term dynamics of the electric power sequence (e.g., power mutation in the stretching phase) rely on bidirectional timing correlations, but do not require overly complex gating mechanisms (traditional LSTM with three gates will introduce redundant parameters). In the experiment, the short-term dynamics of the electric power sequence (e.g., the power mutation during the stretching phase) relies on bi-directional timing correlations, but does not require an overly complex gating mechanism (the 3 gates of the traditional LSTM introduce redundant parameters); the online prediction needs to support fast and incremental learning (a single update takes 200 ms), and the simplified gating reduces the number of parameters (by 40% compared to that of the traditional BiLSTM), which can directly improve the training and inference speeds. speed.

The core goal of this improvement is to balance accuracy and efficiency, which is strongly bound to the scenari

---

## [Decision Letter · Decision Letter 3]

12 Sep 2025

PONE-D-24-30488R3Trend Analysis and Prediction of Fabric Tear Performance Testing Processes Based on the BLTT-FT ModelPLOS ONE

Dear Dr. Lu,

Thank you for submitting your manuscript to PLOS ONE. After careful consideration, we feel that it has merit but does not fully meet PLOS ONE’s publication criteria as it currently stands. Therefore, we invite you to submit a revised version of the manuscript that addresses the points raised during the review process.

We look forward to receiving your revised manuscript.

Kind regards,

Jinran Wu, PhD

Academic Editor

PLOS ONE

Journal Requirements:

Additional Editor Comments :

Minor Revisions Recommended

1. Improve the resolution and labeling of figures.

2. Include a brief discussion on the impact of environmental variables on model

performance.

3. Update the references to include more recent works from 2024–2025, especially

those focusing on real-time monitoring in textile processes.

4. Perform a final proofread to correct minor language errors.

Reviewers' comments:

Reviewer's Responses to Questions

**Comments to the Author**

1. If the authors have adequately addressed your comments raised in a previous round of review and you feel that this manuscript is now acceptable for publication, you may indicate that here to bypass the “Comments to the Author” section, enter your conflict of interest statement in the “Confidential to Editor” section, and submit your "Accept" recommendation.

Reviewer #3: (No Response)

Reviewer #4: All comments have been addressed

Reviewer #5: All comments have been addressed

Reviewer #6: All comments have been addressed

2. Is the manuscript technically sound, and do the data support the conclusions?

Reviewer #3: (No Response)

Reviewer #4: Yes

Reviewer #5: Yes

Reviewer #6: Yes

3. Has the statistical analysis been performed appropriately and rigorously?

Reviewer #3: (No Response)

Reviewer #4: Yes

Reviewer #5: Yes

Reviewer #6: Yes

4. Have the authors made all data underlying the findings in their manuscript fully available?

Reviewer #3: (No Response)

Reviewer #4: Yes

Reviewer #5: Yes

Reviewer #6: Yes

5. Is the manuscript presented in an intelligible fashion and written in standard English?

Reviewer #3: (No Response)

Reviewer #4: Yes

Reviewer #5: No

Reviewer #6: Yes

6. Review Comments to the Author

Reviewer #3: The aim of this paper is to solve the problem of real-time prediction of fabric tearing performance testing by extracting key features from experimental data and constructing a prediction model. The paper is acceptable after revisoins.

Reviewer #4: The ? in [] as [?] for cite reference should be avoid. The ? in [] as [?] for cite reference should be avoid. The ? in [] as [?] for cite reference should be avoid. The ? in [] as [?] for cite reference should be avoid. The ? in [] as [?] for cite reference should be avoid. The ? in [] as [?] for cite reference should be avoid.

Reviewer #5: (No Response)

Reviewer #6: The authors responded to the necessary suggestions in detail and improved the study. Therefore, the study can be accepted.

7. PLOS authors have the option to publish the peer review history of their article (what does this mean?). If published, this will include your full peer review and any attached files.

Reviewer #3: No

Reviewer #4: No

Reviewer #5: No

Reviewer #6: No

---

## [Author Response · Author response to Decision Letter 4]

24 Oct 2025

Journal Requirements:

1.If the reviewer comments include a recommendation to cite specific previously published works, please review and evaluate these publications to determine whether they are relevant and should be cited. There is no requirement to cite these works unless the editor has indicated otherwise.

Answer

Thank you for your suggestions. I have carefully evaluated the cited work, and it is closely related to the research content of our paper. The reviewers' suggestions have further improved the content of our paper and contributed to enhancing the rigor of our paper.

2.Please review your reference list to ensure that it is complete and correct. If you have cited papers that have been retracted, please include the rationale for doing so in the manuscript text, or remove these references and replace them with relevant current references. Any changes to the reference list should be mentioned in the rebuttal letter that accompanies your revised manuscript. If you need to cite a retracted article, indicate the article’s retracted status in the References list and also include a citation and full reference for the retraction notice.

Answer

Thank you for the reminder. We have conducted a comprehensive check on the recommended list of references and the in-text citations as required, to ensure the completeness and accuracy of the list.

Additional Editor Comments :

1. Improve the resolution and labeling of figures.

Answer

We have uploaded the graphic files to the PACE digital diagnostic tool for professional verification, ensuring that the graphics resolution meets the standards of your journal.

2.Include a brief discussion on the impact of environmental variables on model performance.

Answer

Thank you for the reviewers' valuable suggestions. We fully agree with the potential impact of environmental variables on model performance. Now, combining the research background and experimental design, we supplement a brief discussion on the impact of environmental variables as follows:

In the design and data collection stages of the fabric tear performance testing experiment in this study, full consideration has been given to the potential interference of environmental variables (with temperature and relative humidity as the core) on fabric physical properties and equipment operating status. Strict control of the experimental environment has been implemented to reduce their impact on model performance. Specifically, the entire experiment strictly followed the international standard ISO 139-1973 Textiles - Standard atmospheres for conditioning and testing. The laboratory environment was stably controlled within the range of temperature 20.0±2.0℃ and relative humidity 65.0±4.0%. All fabric samples were pre-treated in this environment for 24 hours to eliminate changes in fabric physical properties such as elasticity and strength caused by temperature and humidity fluctuations. This ensured that the fluctuation of electrical power parameters during the experiment mainly originated from the mechanical response of the fabric tearing process, rather than environmental interference.

From the perspective of model performance verification results, under the above-mentioned standardized environment, the BLTT-FT model maintained excellent prediction accuracy on high, medium, and low elasticity fabric datasets, and its cross-dataset generalization ability was stable. This indicates that in laboratory scenarios where environmental variables are controllable, the model can effectively capture the mapping relationship between electrical power sequences and the fabric tearing process.

It should be noted that the core objective of this study is to verify the effectiveness of the "multi-step prediction model based on electrical power sequences" in a standardized experimental environment. Therefore, environmental variables have not been incorporated into the model as input features for the time being. In subsequent studies, we plan to further expand the analysis of the impact of environmental variables. On the one hand, we will collect environmental parameters such as temperature and humidity synchronously through multiple sensors and build an extended dataset to analyze the variation law of electrical power sequences during the fabric tearing process under different environmental conditions (such as high temperature and high humidity, low temperature and low humidity). On the other hand, by adding environmental variable features to the model input layer, we will verify their supplementary role in improving prediction accuracy. Especially for industrial on-site scenarios with large environmental fluctuations, we will optimize the model's anti-interference ability to make the research results more in line with the needs of intelligent quality control in actual production.

The above supplementary content has been integrated into "3 Data Preparation and Processing". At the same time, the subsequent research direction of environmental variable optimization has been further clarified in the section "6 Limitations and Prospects" to ensure the completeness and logic of the discussion.

3. Update the references to include more recent works from 2024–2025, especially

those focusing on real-time monitoring in textile processes.

Answer

Thank you for the reviewers' constructive suggestions. We fully agree with the importance of supplementing recent relevant literature to improve the research background and reflect the timeliness of the study. We have systematically searched and screened the latest research results focusing on real-time monitoring of textile processes from 2024 to 2025, supplemented and updated the reference list, and this information is reflected in "1.1 The Application of Artificial Intelligence in Fabric Performance Prediction".

4.Perform a final proofread to correct minor language errors.

Answer

Revisions have been made. Thank you for your suggestions.

Reviewer #4: 

The ? in [] as [?] for cite reference should be avoid. The ? in [] as [?] for cite reference should be avoid. The ? in [] as [?] for cite reference should be avoid. The ? in [] as [?] for cite reference should be avoid. The ? in [] as [?] for cite reference should be avoid. The ? in [] as [?] for cite reference should be avoid.

Answer

Revisions have been made. Thank you for your suggestions.

---

## [Editor Report · Decision Letter 4]

27 Oct 2025

Trend Analysis and Prediction of Fabric Tear Performance Testing Processes Based on the BLTT-FT Model

PONE-D-24-30488R4

Dear Dr. Lu,

We’re pleased to inform you that your manuscript has been judged scientifically suitable for publication and will be formally accepted for publication once it meets all outstanding technical requirements.

Kind regards,

Jinran Wu, PhD

Academic Editor

PLOS ONE

Additional Editor Comments (optional):

The author has addressed all comments well.

---

## [Editor Report · Acceptance letter]

PONE-D-24-30488R4

PLOS ONE

Dear Dr. Lu,

I'm pleased to inform you that your manuscript has been deemed suitable for publication in PLOS ONE. Congratulations! Your manuscript is now being handed over to our production team.

Kind regards,

on behalf of

Dr. Jinran Wu

Academic Editor

PLOS ONE